# Fairness in link analysis ranking algorithms

## ABSTRACT

In this paper, we investigate the conditions under which minority groups get underrepresented (suppressed) in rankings produced by link analysis ranking algorithms, leading to biased rankings. As recent work shows that link analysis algorithms often prevent minority groups from reaching high rankings, we take a step further in analyzing when do such algorithms amplify pre-existing bias and when can they alleviate it. We find that the most common algorithms using link analysis to create rankings based on nodes' centralities, such as Pagerank and HITS, produce vastly different outcomes: compared to the bias encoded in the degree distribution of a network with multiple communities, Pagerank often mirrors the degree distribution for most of the ranking positions and it can equalize representation of minorities among the top ranked nodes; on the other hand, we find that HITS amplifies pre-existing bias in homophilic networks through a novel theoretical analysis. We find the root cause of bias amplification to be the level of homophily, as well as inequality in the degree distribution. We characterize fundamental differences in how common algorithms may be affected by bias, and explore a series of algorithmic variations in the search for fairness. We find that randomization is a promising tool in debiasing deep inequities encoded in link structures. This work paves the way towards a deep understanding on the difficulty of fixing feature bias in ranking, as the scores that link analysis algorithms output are often used as features in learning-to-rank algorithms, implying that biased features will have a lasting effect on the fairness of many ranking schemes. We illustrate our theoretical analysis on both synthetic and real datasets.

## CCS CONCEPTS

• **Information systems** → **Information retrieval diversity**; **Page and site ranking**; • **Theory of computation** → **Graph algorithms analysis**; *Theory of randomized search heuristics*; **Random network models**; Random walks and Markov chains.

## KEYWORDS

link analysis ranking, search algorithms, information retrieval, pagerank, hits, networks, fairness

**ACM Reference Format:**
Anonymous Author(s). 2018. Fairness in link analysis ranking algorithms. In *Proceedings of Make sure to enter the correct conference title from your rights confirmation emai (Conference acronym 'XX)*. ACM, New York, NY, USA, 16 pages. https://doi.org/XXXXXXX.XXXXXXX

## 1 INTRODUCTION

Ranking algorithms govern the space of information retrieval, with a plethora of research in online search algorithms that can identify relevant and credible sources of information [19, 30]. In the myriad of information sources permeating the web space, filtering through noise in the search for relevant data sources is a difficult task.

A recent line of work has pointed out the potential of such ranking algorithms in amplifying the *perceived* credibility of information sources, particularly when behind the sources are people, groups, or ideologies pertaining to different sensitive attributes (e.g., searching for restaurants owned by minority demographics on Yelp, gender representation on Google search). Whether the bias was reproduced against demographic minority groups [16, 37] or against newer agents that enter the system [13, 25], empirical evidence amounts to a convincing argument that interaction data can amplify popularity among those who are already central in the network. For example, Vlasceanu and Amodio [37] show that even gender neutral queries create disparate representation between men and women on Google search, as searching for the word 'person' shows a disproportionate amount of men in the top results in Google images; in addition, the bias against women in search results correlates with the gender gap index of the country in which the experiment was run from. Another recent example shows that scientific mentorship networks have a vanishing number of women among those with high centrality, much lower than their general proportion [4]; thus, a ranking based on degree would amplify the underrepresentation of women in the field. Thus, ranking heuristics do not necessarily *create* inequality between different demographic groups in the output ranking, but they may reproduce and even *amplify* existing societal inequality.

In this paper, we formally investigate structural causes for which link analysis ranking algorithms amplify, mirror, or reduce inequality between different social groups. We focus on algorithms such as Pagerank [30] and HITS [19] and their variations, and formally study their behavior on a simple yet subtle evolving network model with multiple communities. Our model encapsulates intrinsic inequality through preferential attachment dynamics and homophily, reproducing often observed inequality between the degree distribution of different communities. Such a generative model provides us with a tool for studying the true impact of algorithms on pre-existing inequality. As link analysis ranking algorithms have been developed as alternatives to the degree ranking, we use the degree distribution of different communities in our network as a benchmark, asking the question: *if the degree ranking is a baseline for social capital inequality, when and why do link analysis algorithms induce an even more unequal distribution in the ranking scores, and when do they correct degree bias?*

The answer to these questions reveals subtle dynamics: different algorithmic choices and levels of inequality have a drastically different impact on the fairness of outcome rankings. We show that the interplay of high homophily and unequal degree distribution plays a central role in amplifying bias in rankings produced by HITS,

using a novel theoretical analysis based on a model of evolving networks with multiple communities and bias in the degree distribution, and validating it on real-world data. On the other hand, we find that Pagerank corrects the degree bias for top ranked nodes, confirming a recent theory [3]; for most other nodes, we show that it closely follows the bias in the degree distribution in both synthetic and real data, illustrating limitations of recent theories on its impact on fairness. Our analysis paves the way for understanding fundamental differences in the impact of algorithmic choices on bias amplification. Indeed, we find that our results provide a first theoretical explanation: on the one hand, the difference between Pagerank and HITS can be explained through a theory of random walks, showing that it is the reinforcing effect of backward-forward paths that HITS employs (as opposed to just forward walks in Pagerank) that greatly amplifies bias in an already clustered network; on the other hand, random restarts in the random walks together with normalizing the influence of high indegree nodes provide a limited help in theory, as we show that it brings a ranking closer to the degree ranking (and therefore mirroring the degree bias); in practice, we find randomization to improve fairness to a greater extent, as we show on real data.

*Our contributions:*

- Using a model of network growth that encodes bias in the degree distribution, we find evidence that Pagerank reproduces the bias in the degree distribution, except for the top ranked nodes, for which it alleviates it. (Section 3)
- Inspired by this finding, we develop theory that predicts conditions for which bias is alleviated, reproduced, or amplified in link analysis algorithms. Through a novel theoretical argument, we find that homophily is the main predictor in amplifying degree bias in HITS, deriving a closed-form condition showing that the level of bias amplification directly depends on how homophilic the network is—more clustered networks lead to a more biased ranking. To our knowledge, our results are the first to highlight, based on a theory of random walks, fundamental differences between the impact on fairness of these two algorithms. (Section 3)
- We find evidence to support our findings in real-world datasets with varying levels of homophily. In practice, we find that Pagerank can be less fair than predicted by theory. (Section 4)
- We characterize the role of randomization in improving fairness, providing empirical and theoretical evidence that randomization in HITS improves fairness in HITS, essentially mirroring the degree bias or even improving upon it. We experiment with other variations of HITS based on multiple eigendirections, finding that fairness very much depends on the number of dimensions used and on the specific data. (Section 5)

These results open several avenues of research, as we find fundamental differences among different algorithms that use network structures. We conclude by highlighting a series of proposals in how to progress in fair ranking using algorithmic design choices sensitive to the root cause of bias—whether that be structural (in the form of centrality differences) or social (in the form of homophilic behavior).

## 2 BACKGROUND AND RELATED WORK

### 2.1 Background and modeling choices

*Link analysis algorithms.* Ranking algorithms that leverage the connections formed between different information sources (websites, blogs, etc) were developed on the assumption that a link is equal to an endorsement. Popular sources gain their credibility through many links pointing towards them, weighted by different functions that capture how important downstream paths are. In this paper, we focus on two main algorithms, Pagerank [30] and Hyperlink-Induced Topic Search (HITS) [19], described below:

*The Pagerank algorithm* was developed to leverage random walks on graphs, with the intuition that the more these walks land on a node, the more important or central that node is [30]. Closely related to the use of degree centrality, the asymptotic behavior of these random walks governs their ranking position. Indeed, for undirected graphs, the stationary distribution exists and is proportional to the degree distribution. For directed graphs, the stationary distribution does not necessarily exist without a small modification, that of adding a random re-start, giving us the Pagerank equation:

$$\mathbf{x}_{t+1} = \eta \cdot P \cdot \mathbf{x}_t + (1 - \eta) \cdot \mathbf{v}, \tag{1}$$

where $P$ is the transition matrix, $1 - \eta$ is the restart probability, and $\mathbf{v}$ is the vector of teleportation (taken to have all coordinates equal to $1/n$, where $n$ is the number of nodes in the network). For a graph $G(V, E)$ with a set of nodes $V$ and a set of edges $E$, denote its adjacency matrix by $A$ and note that $P_{ij} = \frac{A_{ij}}{\sum_i A_{ij}}$ ($\sum_i A_{ij}$ is the outdegree of a node $j$). Equation 1 is proven to have a stationary distribution $x_{t+1} \rightarrow_t x^*$, which is the coefficient vector denoting the importance of each node.

*The HITS algorithm* was developed to account not only for those of high indegree [19], but also to provide a sense of credibility by accounting for 'hubs' (nodes of large outdegree). In doing so, what matters most in the HITS algorithm is whether one's neighbors are trusted sources of information. This trust is formalized through being a 'hub' where information aggregates, contrasted with an 'authority' to which many hubs point to. The algorithm formalizes this through a bipartite transformation of a graph, where each node $u \in V$ has now a hub score $h(u)$ and an authority score $a(u)$ that reinforce each other at every time step $t$ through the update equations:

$$a^{(t+1)}(u) = \sum_{v,(u,v) \in E} h^{(t)}(v) \text{ and } h^{(t)}(u) = \sum_{v,(u,v) \in E} a^{(t)}(v) \tag{2}$$

This set of equations is proved to converge to $a^*$ and $h^*$, respectively, and in fact, to show that the hub and authority scores are the principal eigenvectors of a variant of the adjacency:

**THEOREM 2.1 (KLEINBERG, 1999).** *$a^*$ is the principal eigenvector of $A^T A$ and $h^*$ is the principal eigenvector of $A A^T$.*

In theoretically analyzing these algorithms, we employ a network model, called the Biased Preferential Attachment Model (BPAM), that reproduces commonly observed characteristics of biased networks: multiple communities, homophily, and a skewed degree distribution that induces a power-law distribution with *different* coefficients for different communities. In such a model, the degree ranking is inherently unequal: as we move towards higher ranks,

the proportion of a minority group (with a lower coefficient in the power law degree distribution) effectively vanishes, creating a so-called 'glass ceiling effect' as measured through social capital [4]. We employ this terminology to investigate when a glass ceiling effect gets *reduced* or *amplified* in the ranking produced by link rank analysis algorithms, i.e. when does a minority get better or worse access to higher rankings. We find a continuous relationship between increased homophily and bias amplification in HITS in the BPAM through a mean-field analysis employing a theory of random walks.

## 2.2 Related work in fairness in ranking

*Fairness in link analysis ranking.* The question of fairness in link analysis algorithms has only recently started to gain attention. Espín-Noboa et al. [16] analyze Pagerank under network models similar to ours, yet they only define fairness with respect to statistical parity, without comparing to the degree distribution. Our work brings a novel analysis by characterizing when bias is *amplified* as compared to pre-existing inequality. Antunes et al. [3] theoretically show that, in network models with homophily and a power-law degree distribution with different coefficients among different groups, the Pagerank score distribution for top nodes follows a power law with the same coefficient for different communities; yet, this result does not capture the distribution of any other node that is not in tail of the degree distribution to begin with. In our work, we find new evidence that Pagerank only improves the ranking of minorities as compared to degree for the very top at best in synthetic data (Section 3), and may in fact entirely mirror the degree bias in real data (Section 4). This result motivates us to understand the fundamental ways in which structural bias permeates link analysis.

On the other hand, HITS has received much less scrutiny with regard to general fairness questions, however with significant research done on its effectiveness (e.g. Najork et al. [27] showing that it can outperform Pagerank on the Web in terms of the NDCG score and the mean average precision). Previous empirical evidence has shown that completely disconnected components will have an odd effect on HITS, with larger groups being promoted [9, 22, 29]—an effect known as the *tightly knit community* (TKC) effect. Our results present a formal and more in-depth analysis of the TKC effect: we find that groups do *not* have to be completely disconnected for minorities to suffer in the ranking; instead, there is a continuous link between homophily and minority underrepresentation in HITS. We explain this link through a theory of random walks, noting that high indegree nodes reinforce each other's authority scores. We show that renormalizing the influence of high indegree nodes and adding a random restart (equivalent to randomized HITS [29] and conceptually similar to the SALSA algorithm [22]) alleviates the bias amplification, but only inasmuch as following the degree distribution. In previous work, randomized HITS [29] and SALSA [22] show an empirical diversity-enhancing effect, but only for the top $5 - 10$ ranks. We investigate in the potential of such variations in improving fairness at all ranks—as opposed to just the first few—in Section 5. To our knowledge, our work is the first to explain a generalized theory of bias in link analysis ranking algorithms, finding fundamental differences between algorithmic choices on the impact of bias amplification. We conclude with an empirical analysis of other variations of HITS that use multiple dimensions

in the eigenspace, motivated by stability results on the use of multiple eigenvectors [29] and by recent work showing that one extra dimension can improve fairness in data representations through PCA [31]. We find that, empirically, using multiple eigenvectors does not present a consistent path for fairness improvement, as it very much depends on number of eigenvectors and the data used. This analysis opens an avenue of research for finding a systematic pattern in which lower dimensions may improve fairness by better representing minorities in a higher-dimensional embedding.

*Fairness in learning-to-rank and machine learning.* Fairness issues pertaining to the representation of minority groups in rankings become particularly important when centrality metrics (such as Pagerank centrality and HITS authorities) are used as features in learning-to-rank algorithms [1, 14, 24, 36]. Drawing from a vast literature on fairness in supervised machine learning [2, 15, 17, 18, 20, 38], fairness constraints in the form of statistical parity have been proposed in different learning-to-rank procedures [7, 8, 11, 32, 39, 40]. Such constrains often come as a post- or in-processing technique, without modeling the generative process of feature distribution (for an extensive survey on fairness in learning-to-rank and related tasks, see [41] and [42]). In this work, we tackle the way bias permeates the feature space, with a focus on link analysis ranking algorithms whose outcomes often get used as relevance features. On its own, the problem of fairness in link analysis is current and presents subtle challenges; beyond that, it has repercussions on the impact of feature bias on machine learning algorithms used in rankings. We argue that using evolving network models in link analysis ranking presents us with a unique opportunity, absent in the learning-to-rank literature: modeling bias created through evolving network models can give us a unique tool to understand the particular structures that contribute to it, such as group inter-connectivity, dynamics of connections that create unequal degree distributions etc.

## 3 A THEORY OF BIAS IN HITS

## 3.1 Model description and preliminaries

We first start by analyzing the Pagerank and HITS algorithms on synthetic data generated from a model of network evolution that encodes pre-existing bias in the degree distribution, which we describe below. We follow with a theory predicting the conditions in which bias gets overly amplified in HITS, finding a closed-form relation involving homophily.

*The Biased Preferential Attachment Model.* is a variant of the Preferential Attachment Model that has been recently proposed [4], leading to the creation of multiple communities and a clustering effect. For the simple case of two communities (a community here might mean a political affiliation, a demographic associated to the search items etc), we assume that each node has a label, blue (B) or red (R), and at each point in time the network grows as:

- *Minority-majority partition*: a new node $u$ enters the network and receives the label $R$ with probability $r$ and the label $B$ with probability $1 - r$. We assume that the red community is in minority, with $0 \leq r \leq 1/2$.
- *Preferential attachment (rich-get-richer)*: $u$ chooses a node uniformly at random and *copies* one of its edges. This is equivalent to the new node connecting proportionally to the

ending node's degree, $\mathbb{P}(v \text{ is chosen}) = d_t(v)/\sum_{u \in V_t} d_t(u)$, where $d_t(x)$ denotes the degree of node $x$ at time $t$, and $V_t$ is the set of nodes in the graph at time $t$. Preferential attachment is what leads to a *rich-get-richer* effect, where a few nodes are very well-connected and most nodes have very few connections—an effect observed in many online networks, such as the Web structure [5, 6].

- *Homophily:* if the new node has a different label than the node it chooses to connect to, the connection is accepted with probability $\rho$ and the process is repeated until an edge is formed. A value of $\rho$ closer to 0 means a more homophilic (and therefore segregated) network, while a value of $\rho$ closer to 1 means a more integrated network. The homophily parameter $0 \leq \rho \leq 1$ captures the fact that a person less similar is less likely to be eventually chosen than one of the same kind. Homophilic behavior has been observed in many real networks among different attributes (e.g., demographics, location, interest groups, professional collaborations) [10, 26].

Thus, as the network grows according to this model, exactly one node and one directed edge are added at each timestep. In this model, everyone has an out-degree of 1 and a different in-degree. We repeat this model $d$ times until everyone has an out-degree of $d$, for a choice of $d$. Just as in the Preferential Attachment Model, this model asymptotically leads to a power law distribution for each of the groups, with a different coefficient for each community [4]:

THEOREM 3.1 (AVIN ET AL, 2015). *A graph sequence $G(n)$ generated through the Biased Preferential Attachment Model exhibits a power law degree distribution asymptotically:*

$$top_k(B) \sim k^{-\beta(B)},$$
$$top_k(R) \sim k^{-\beta(R)}, \tag{3}$$

*with $\beta(B) < 3 < \beta(R)$, where $top_k(R)$ and $top_k(B)$ denote the number of red and blue nodes with a degree of at least $k$, respectively.*

These coefficients can be analytically computed from the model. When there is no homophily (the network is completely integrated), complete homophily (the network is completely segregated), or the two subgroups are equal in proportion, the coefficients of the power law for the degree distribution are also equal. For intermediate values of homophily and in the presence of a minority group, the coefficients will indeed be different, leading to a so-called 'glass ceiling effect' [4]: minority nodes have a vanishing fraction in the top degrees.

## 3.2 Synthetic data analysis using BPAM

We simulate the BPAM for $n = 1,000$ nodes and outdegree $d = 6$. For each network instance simulated from this model, we compute the degree distribution, the Pagerank, and the HITS authority scores and we plot the ratio of minority members among those in the top $x$ percentage of ranking and above on the $x$-axis (plotted in log-scale), and adding a dashed horizontal line for the population minority ratio. All simulations are averaged over $1,000$ iterations. For example, the leftmost plotted point in Figure 1 is the ratio of minority members among the entire population, top 100% (so, top

1,000 people). In a sense, the dashed line is our fair baseline, also equivalent to achieving statistical parity: the further away the minority ratio plotted is from the dashed (often smaller), the less fair the ranking becomes. We know from Theorem 3.1 that the proportion of minority members among top ranked people by degree goes to 0, an illustration of what a glass ceiling effect may look like in practice. We note that for $r = 0.5$ or $\rho = 1$, the degree distribution is the same among the two communities, and then the Pagerank and HITS distributions are also fair (as in, are equal to or very close to the proportion of minority nodes in the population represented by the black dashed line). The closest $\rho$ gets to 0 (Figure 1 a), the two communities are getting increasingly disconnected, and we note that HITS becomes progressively more unfair, while degree and Pagerank become more fair. For example, while we have 30% of minority proportion in the population, there are under 20% among those in top 10% and all above ranks in HITS. For moderate homophily (e.g. $\rho$ equal to 0.3 and 0.5 in panels b and c), HITS and degree ranking are quite similar, both leading to a vanishing minority fraction among the top ranked. For Pagerank, while finding novel evidence that it is similar to the degree ranking for most of the ranking, we also find improvement in the minority ratio for the very top ranked nodes (note that Antunes et al. [3] proves that Pagerank is fair only for the very top nodes). This shows that Pagerank and HITS do not generally introduce bias when the degree is unbiased, but rather reproduce the bias that is already existing. Yet, they differ in how much of that bias they reproduce, with HITS being particularly sensitive to the level of homophily in the network.

## 3.3 Bias in HITS: a Mean-Field Analysis

We complement our experimental results with a theoretical analysis of the HITS algorithm to elucidate the structural reasons behind the amplification of bias in the rankings that HITS produces. Specifically, the BPAM helps accurately expound the role of homophilyAs noted in Section 2, Borodin et al. [9], Lempel and Moran [22], Ng et al. [29] argue that the TKC effect reduces the diversity of search results in the top $5 - 10$ queries, and conjecture that this effect emerges when communities are completely disconnected in the larger network. We find a more subtle effect: even in connected networks, HITS can amplify the underrepresentation of minorities in the ranking outputted as compared to the degree ranking, and moreover, the more homophilic the network is, the more pronounced the bias amplification.

THEOREM 3.2. *For a network $G(V, E)$ drawn from BPAM with $N$ nodes, two communities, minority ratio $r$, and homophily parameter $\rho$, the following hold:*

(1) *the following inequalities are true for the authority scores $a(\cdot)$ obtained from the HITS algorithm:*

$$\overline{a(R)} \leq \overline{a(B)} \text{ and } \overline{a(R,k)} \leq \overline{a(B,k)} \tag{4}$$

*for $0 \leq r \leq 0.5$ and $0 \leq \rho \leq 1$, where $\overline{a(C)}$ (and $\overline{a(C,k)}$, respectively) denotes the average authority score of community of color $C$ (and degree class $k$, respectively).*

(2) *for nodes $u \in R$, $v \in B$ of similar degree , the ratio $\frac{a^{(t)}(u \in R)}{a^{(t)}(v \in B)}$ is an increasing function in the homophily parameter $\rho$ for*

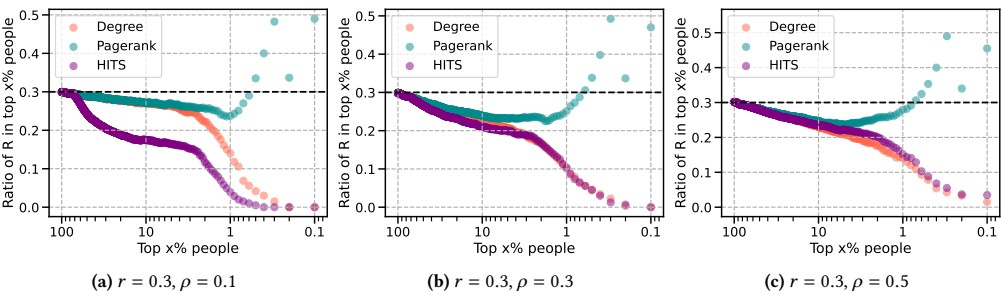

**(a)** $r = 0.3, \rho = 0.1$        **(b)** $r = 0.3, \rho = 0.3$        **(c)** $r = 0.3, \rho = 0.5$

**Figure 1: Representation of the minority group R in the ranking of the nodes based on degree (orange), HITS (purple), and Pagerank (blue), for directed networks simulated from the BPAM with $n = 1,000$ nodes and outdegree $d = 6$.**

any $t \geq 2$, where $a^{(t)}(\cdot)$ denotes the authority score of a node after $t$ iterations of the HITS algorithm.

The proof of this theorem relies on the properties of the Biased Preferential Attachment Model as well as on being able to interpret HITS in terms of random walks on graphs through a mean-field analysis. We provide a sketch of the proof in this section, with full proofs for all intermediate steps found in the Appendix 7.2.

The crux of the proof relies on using the model to compute an approximation for the the authority score of nodes as a product of the nodes' indegree and an additional factor, termed here as a multiplicative factor. By analyzing the behavior of the multiplicative factor as a function of the model parameters, we can investigate how the authority score distribution compares to the degree distribution. First, we note that the authority score of the nodes after $t$ iterations of the HITS algorithm is proportional to the number of paths that alternate between backward and forward direction of the edges, starting from each node [9] (initializing with hub scores equal to 1, then performing a first update on the authority scores as per equation (2)). We can then approximate the number of backward-forward paths for nodes belonging to each community using the properties of the model. We detail our approximation in the Appendix, noting that it boils down to identifying a dominant term in a recurrent equation over the iterations $t$. We denote the approximation of the HITS at iteration $t$ by:

$$a^{(t)}(v \in R) \approx d^{in}(v) \cdot (d-1) \cdot MF^{(t)}(R),$$
$$a^{(t)}(v \in B) \approx d^{in}(v) \cdot (d-1) \cdot MF^{(t)}(B),$$
(5)

where $d^{in}(v)$ is the indegree of a vertex $v$ (when $d^{in}(v) is O(1)$ in $N$), $d$ is the (constant) outdegree of a vertex, and $MF^{(t)}(R)$ and $MF^{(t)}(B)$ are the multiplicative factors that allow to compare the HITS ranking with the degree ranking. For top nodes, a similar analysis follows (details in the Appendix 7.2). To prove the first (1) part of the theorem, we show the following results (proof in Appendix 7.2):

**Proposition 3.3.** For any $t \geq 2$, $MF^{(t)}(B) \geq MF^{(t)}(R)$, for all $0 \leq r \leq 0.5$ and $0 \leq \rho \leq 1$.

Then, knowing that the average indegree of a red node is lower or equal to the average indegree of a blue node (Theorem 4.1, part 1 in Avin et al. [4]), we get that by averaging over the set of red and

blue nodes, respectively, $\overline{a(R)} \leq \overline{a(B)}$. Similarly, averaging over degree classes, (all red nodes and all blue nodes of indegree equal to $k$, respectively, for all $k$), we get that $\overline{a(R,k)} \leq \overline{a(B,k)}$, which proves the first part of the theorem.

To prove part (2) of the theorem, we consider the multiplicative factors as functions of the homophily parameter $\rho$ and define $F^{(t)}(\rho) := \frac{MF^{(t)}(R,\rho)}{MF^{(t)}(B,\rho)}$ (in fact, we detail in the Appendix 7.2 that $MF^{(t)}(R, \rho)$ and $MF^{(t)}(B, \rho)$ contain the same term that depends on $t$, so $F^{(t)}(\rho) := F(\rho)$). We note that for nodes of similar indegree, the ratio of their authority scores boils down to the ratio of the multiplicative factors in the approximation. Thus, we show the following:

**Proposition 3.4.** Define $F(x) := \frac{MF^{(t)}(R,x)}{MF^{(t)}(B,x)}$ as a function of the homophily parameter $\rho \in (0, 1]$. Then, $F(x)$ is an increasing function in $(0, 1]$ with $F(1) = 1$.

*Intuition:* The intuition that stems from this analysis is as follows: $F$ computes the ratio of the multiplicative factor of red and blue nodes in determining their authority score. When $F(x) = 1$, it implies that both communities have the same multiplicative factor, rendering the HITS ranking the same as the degree ranking. That means that nodes of similar degree will have a similar chance of showing up in a ranking of $k$ nodes regardless of color, and thus, by aggregation, the ratio of R nodes in the top k in the HITS ranking will be the same as in the degree ranking. When $F \neq 1$, that means that one community is 'bumped up' (or 'bumped down') more than the other, which will change the ranking. For example, if the red community gets a discount factor ($MF^{(t)}(R) < 1$) and the blue community gets a boosting factor ($MF^{(t)}(B) > 1$), that means that the red community is 'pushed down' in the ranking as more blue nodes are overtaking red nodes that initially had similar or even better degree in the HITS ranking. If both communities get either a discount or a boosting factor, whichever factor is higher will 'boost' that community upper in the ranking as compared to the original degree ranking. Thus, if the red community consistently has a lower multiplication factor than the blue community, that means that they are consistently 'pushed down' in the HITS ranking as compared to the degree distribution. Proposition 3.4 essentially shows that more homophilic networks experience exacerbated bias against minority (red) nodes compared to the initial degree distribution,

while integrated communities will mirror the bias encapsulated by the degree distribution.

The fundamental difference between HITS and Pagerank lies in the way nodes aggregate centrality: while for Pagerank, the transition matrix is equivalent to taking random walks in the 'forward' direction starting from a node (following its outdegree), with a random restart, for HITS, the update equations are equivalent to taking 'backward-forward' paths starting from a node, without any restart. The outdegree is constant in our model and the random restart in Pagerank serves in discounting paths with exponential decay in the restart probability (for a full theoretical analysis of Pagerank, see Antunes et al. [3]). On the other hand, the backward-forward paths mean that nodes with high indegree are greatly advantaged, amplifying their authority from other high indegree nodes with whom they have a commoon neighbor that points to both (in branching processes terminology, a common 'child'). These paths are not discounted and serve to amplify the scores of already well-connected nodes, who are likely to connect with others of the same color due to homophily.

## 4 EXPERIMENTAL EVIDENCE OF BIAS

We worked with several network datasets that contain directed edges, varying levels of homophily, and two main communities: a majority and a minority. In each of these datasets, either the majority or the minority community has the 'advantage' in the degree distribution (meaning that they are over-represented at the top of the degree distribution, as the glass ceiling definition formalized in Avin et al. [4]). The datasets are described below, color-coded by which group has the degree advantage (blue for majority, red for minority).

(1) **APS:** The APS citation network [21] contains 1, 281 nodes, representing papers written in two main topics: Classical Statistical Mechanics (CSM), constituting 37.5% of the papers, and Quantum Statistical Mechanics (QSM), accounting for the rest of 62.5% of the papers. As Lee et al. [21] analyze, the dataset has high homophily, meaning that each subfield cites more papers in their own field than in the other field.

(2) **DBLP:** The DBLP citation dataset of computer scientists was collected from the DBLP platform [23, 33, 35], an online database that records most publications in computer science. We use version V10 of the dataset, using the authors present in all papers as nodes and associating a perceived gender of each node through the genderize.io API; we retain the nodes for which the probability of a gender is over 90%. A directed edge is created between two authors if a paper by the first author cites a paper by the second author. We extract the largest weakly connected component, obtaining a graph with 1, 224, 996 nodes and two communities, men (88%) and women (22%). The data has low homophily.

(3) **Instagram:** An interaction network from Instagram collected by Stoica et al. [34] containing 553, 628 nodes and 652, 931 edges, where everyone has a labeled gender (45.57% men and 54.43% women). Each edge between two users represents a 'like' or 'comment' that one user gave another on a posted photo. The data has moderate homophily.

Table 1 presents data characteristics, including the number of nodes, edges, minority percentage, and an estimated homophily

### Table 1: Data characteristics.

|  | Nodes | % minority | Edges | HRI |
|---|---|---|---|---|
| APS | 1,281 | 31.7 % | 3,064 | 0.12 |
| DBLP | 1,224,996 | 22 % | 95,160,219 | 0.74 |
| Instagram | 539,023 | 45.6 % | 640,211 | 0.44 |

factor. Each network either has an advantaged majority group or an advantaged minority group, illustrated in Figure 2: when the complementary cumulative distribution function (CCDF), plotted in log-log scale, of a group is higher than the other on an interval, then the representation of that group for the degree classes belonging to that interval is higher than for the other group. When that interval includes the top of the degree hierarchy, it is used as evidence to illustrate a glass ceiling effect against one group.

In order to assess whether a network is homophilic or not, we use the **homophily rarefaction index** (HRI) [43], which computes the fraction of cross-community edges over the expected number of cross-community edges that the BPAM gives, which is $2 \cdot r \cdot (1 - r) \cdot |E|$, where $E$ is the set of edges in the network. Other methods are the Newman assortativity index [28] and the asymmetric homophily index [21]. An HRI closer to 1 means a more integrated dataset, while a lower HRI indicates higher homophily. The APS and DBLP datasets both have a majority group that has an advantage in the degree distribution, with APS being the most homophilic and DBLP the least. The Instagram dataset has a minority group that has a degree advantage and moderate homophily.

We tested Pagerank and HITS on all three datasets, illustrating the ratio of the minority group among each rank in Figure 3. We notice that the data captures the dynamics predicted by the BPAM: as APS and DBLP have an advantaged majority group, the minority group gets underrepresented in the degree, and even more so in HITS. For APS, HITS (purple) amplifies the under-representation of the minority the most as compared to the degree distribution (orange), which consistent with the fact that it is the most homophilic network. DBLP has the least homophily, consistent with HITS slightly amplifying the bias seen in the degree hierarchy. Instagram, on the other hand, has an advantaged minority group by degree, which is also over-represented in the ranking produced by both Pagerank and HITS for most ranks. Instagram is more homophilic than DBLP but less than APS, explaining why HITS amplifies more of the minority advantage in Instagram than it amplifies the majority advantage in DBLP, and less than in APS.

These experiments show that the BPAM is relatively accurate in reproducing the data behavior on ranking algorithms. Furthermore, our experiments reveal the subtle effect of homophily, showing that in more homophilic networks, HITS amplifies bias against minorities. We note that while Pagerank is provably fair [3], it actually reproduces the degree distribution bias on DBLP, while preserving statistical parity for APS and Instagram for most of the ranking (except the very top, where it inherits some bias as it departs from the statistical parity dashed line). This points to a cause of inequality stemming from the *outdegrees* this time (as they are the normalizing factor of Pagerank, and are not constant in the read data like the BPAM assumes), prompting further future investigations.

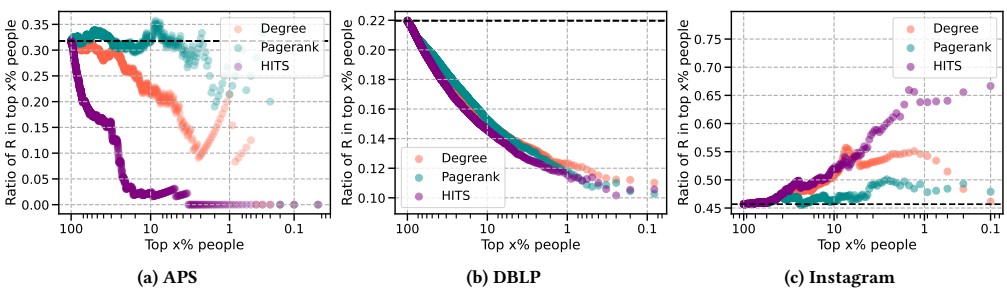

**Figure 2: CCDF of the degree distribution of the minority group (red) and the majority group (blue) for the real-world networks.**

**Figure 3: Representation of the minority group R in the ranking of the nodes based on degree (orange), HITS (purple), and Pagerank (blue), for the real-world networks.**

## 5 RANDOMIZATION: A PATH TO FAIRNESS?

Our findings of the structural differences between Pagerank and HITS motivate our quest for finding algorithmic variations that can improve observed bias. We have showed that HITS amplifies the bias in the degree distribution due to its reinforcement of authority scores from high indegrees (through the backward-forward path dynamics). Thus, a natural question arises: can we debias HITS by normalizing the indegree influence, and would a restart in the random walk (similar to Pagerank) additionally help? We are in luck, as such a variation, named randomized HITS [29] has been shown to have perform well in terms of query relevance, yet has not been analyzed with respect to fairness to different communities.

*Randomized HITS:.* we introduce a random restart probability, similar to the restart probability in Pagerank, where with some probability $\epsilon$ a random surfer resets and chooses a node uniformly at random. Conversely, with probability $1 - \epsilon$, it adheres to the backward-forward iterations inherent to HITS, yet with a normalization. We choose the restart parameter value to be equal to $\epsilon = 0.15$, just like for the implementation of Pagerank in our analysis. We note that this variation has been proposed in Section 5.1 of Ng et al. [29], formalized through an iterative process as:

$$a^{(t+1)} = \epsilon \cdot \overrightarrow{\mathbb{1}} + (1 - \epsilon) \cdot A_{row}^T \cdot h^{(t)},$$
$$h^{(t+1)} = \epsilon \cdot \overrightarrow{\mathbb{1}} + (1 - \epsilon) \cdot A_{col} \cdot a^{(t+1)}, \quad (6)$$

where $\overrightarrow{\mathbb{1}}$ is the all-ones vector, $A_{row}$ and $A_{col}$ are the row- and column-stochastic versions of the adjacency matrix $A$, respectively.

We note that randomized HITS is conceptually similar to the SALSA algorithm [22], for we find very similar results, omitted in this article due to space constraints. We show that randomized HITS essentially reproduced the degree bias in theory, yet, in practice, it may even alleviate such bias.

**Proposition 5.1.** *For a network $G(V, E)$ drawn from BPAM with $N$ nodes, two communities, minority ratio $r$, and homophily parameter $\rho$, the authority scores produced by randomized HITS can be approximated by the indegree distribution, scaled by a coefficient:*

$$a^{(t)} \approx \epsilon \cdot \overrightarrow{\mathbb{1}} + \mathbf{d}^{in} \cdot A(\epsilon, d, t),$$

*where $\mathbf{d}^{in}$ is the vector of indegrees and $A(\epsilon, d, t)$ is a constant in $t$.*

The proof is found in the Appendix 7.3. Synthetic data generated from the BPAM with various parameters validate our claim (Figure 4). Real data shows a more promising story: randomized HITS is consistently the fairest algorithm, bringing the minority ratio among those above a certain rank closest to the population minority ratio, for most ranks (Figure 5). Randomized HITS performs similarly to Pagerank, except for DBLP, for which Pagerank is actually quite similar to the degree ranking, but randomized HITS improves the minority representation as compared to the degree ranking (for example, for DBLP, for the top 10% of the ranks, approximately 16% of them are women in the degree ranking, compared to 21% in randomized HITS, which is close to the population women ratio of 22%).

*Subspace HITS:.* Finally, we experimentally explore the potential multiple dimensions in the eigenspace to improve fairness. We combine multiple eigenvectors of the $A^T A$ matrix for computing the

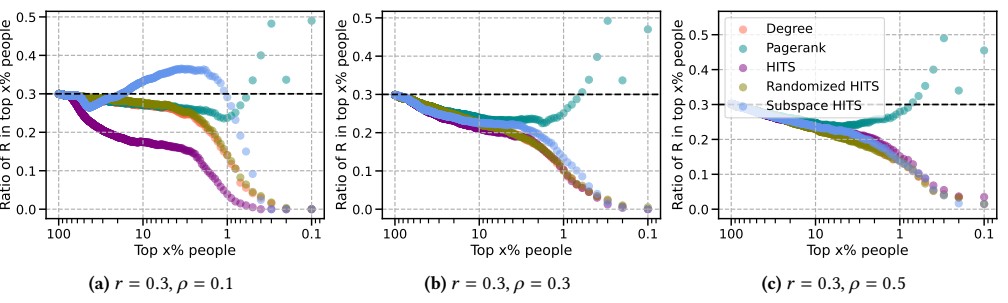

**(a)** $r = 0.3, \rho = 0.1$   **(b)** $r = 0.3, \rho = 0.3$   **(c)** $r = 0.3, \rho = 0.5$

**Figure 4: Representation of the minority group R in the ranking of the nodes based on degree (orange), HITS (purple), randomized HITS (olive), and Pagerank (blue), for directed networks simulated from the BPAM with $n = 1,000$ nodes and outdegree $d = 6$.**

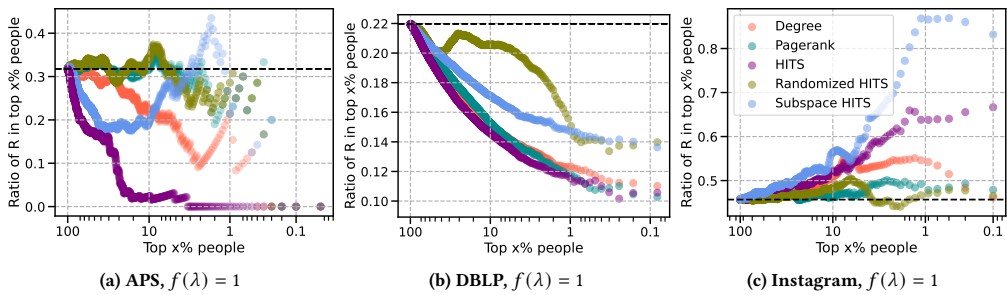

**(a) APS,** $f(\lambda) = 1$   **(b) DBLP,** $f(\lambda) = 1$   **(c) Instagram,** $f(\lambda) = 1$

**Figure 5: Representation of the minority group R in the ranking of the nodes based on degree (orange), HITS (purple), and Pagerank (blue), using $6$ eigenvectors and $f(\lambda) = 1$, for the APS (a), DBLP (b), and Instagram (c) datasets.**

authority score, instead of just using the principal eigenvector. The intuition is that more information about different communities may be stored in lower eigenvectors (for example, it has been shown that one extra dimension in PCA greatly improves the fairness of representation of minority groups in the dimensions chosen for projection [31]). We note that this variation has been proposed in Section 5.1 of Ng et al. [29], formalized as:

(1) Choose the first $k$ eigenvectors $v_1, v_2, \cdots, v_k$ of $A^T A$ with their corresponding eigenvalues $\lambda_1, \lambda_2, \cdots, \lambda_k$. We choose $k$ between 6 and 10 in experiments.

(2) Compute the authority of node $j$ as $a_j = \sum_{i=1}^{k} f(\lambda_i)(e_j^T v_i)^2$, where $e_j$ is the $j$-th basis vector and $f(\cdot)$ is a function of our choice (we experiment with $f(\lambda_i) = 1$ and $f(\lambda_i) = \lambda_i^2$).

However, subspace HITS is not as stable, nor is it as fair, as it sometimes is less fair than the degree ranking (for APS except the very top, and Instagram all throughout), and sometimes more fair the degree ranking (for DBLP), in Figure 5. Synthetic data shows a similar behavior, with high homophily showing some improvement over degree, whereas moderate degree mostly reproducing the degree bias (Figure 4). For the interested reader, we experiment with choosing different number of eigenvectors in the Appendix 7.4, noting varying levels of fairness. This opens a research path for investigating the optimal number of dimensions and their choice in improving bias.

## 6 CONCLUSIONS AND FUTURE DIRECTIONS

In this paper, we provide an in-depth analysis of how deeply bias permeates network algorithms used in link analysis ranking. By employing a model of biased networks—which encodes bias at the level of the degree distribution across various communities—we uncover profound connections between degree bias, homophily, and link analysis algorithms. This is the case even when strategies like randomization or deploying lower dimensions in the eigenspace embedding of data are applied. As we formally show the role of homophily in amplifying bias in link analysis ranking algorithms such as HITS, we find nuance and formalism to the previously observed empirical effect called *tightly knit community* effect.

We find several promising directions for future work: first, an understanding of how randomized HITS may behave on different models of network, given our optimistic empirical results from real datasets. Second, a future direction could delve deeper into the theory of network embeddings to comprehend the extent of structural bias. Specifically, is the HITS authority matrix $A^T A$ indicative of potential bias amplification issues in other embedding types, and are higher dimensional embeddings beneficial in recovering the lost representation of a minority group? Third, future work should delve into understanding the impact of interventions on the behavior or dynamics in subsequent timesteps: how would network dynamics change in response to users viewing more fair rankings, and thus, what is the long-term impact of fairness in link analysis ranking?

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

## 7 APPENDIX

### 7.1 Asymptotic degree distribution in BPAM

We start by computing two types of probabilities, for a node $v \in V$ and a color $C \in \{R, B\}$:

$$\mathbb{P}(v \text{ connects to a node of color } C) \tag{7}$$

and

$$\mathbb{P}(v \text{ has connections from a node of color } C). \tag{8}$$

Note that the probabilities in equations (7) and (8) are different. We compute these probabilities asymptotically as the size of the network $N$ grows to infinity. Denote by $d_{tot}^N(C)$ the total degree of nodes of color $C$. Recall that the total degree in the network of size $N$, is $2Nd$. As in Avin et al. [4], denote by $\alpha_N := \frac{d_{tot}(R)}{2Nd}$. It follows from Lemmas 4.4 and 4.5 in Avin et al. [4] that $\lim_{N \to \infty} \alpha_N = \alpha < r$, so the fraction of edges with red node as a target is smaller than the fraction of red nodes. In Avin et al. [4] this is called the *power inequality*. Then, compute the probabilities (7) in the asymptotic regime as:

$$p_{RR}^{out} := \mathbb{P}(v \text{ connects to a node of color } R | v \in R) = \frac{\alpha}{\alpha + \rho(1 - \alpha)},$$

$$p_{RB}^{out} := \mathbb{P}(v \text{ connects to a node of color } B | v \in R) = \frac{\rho(1 - \alpha)}{\alpha + \rho(1 - \alpha)},$$

$$p_{BR}^{out} := \mathbb{P}(v \text{ connects to a node of color } R | v \in B) = \frac{\rho\alpha}{\rho\alpha + 1 - \alpha},$$

$$p_{BB}^{out} := \mathbb{P}(v \text{ connects to a node of color } B | v \in B) = \frac{1 - \alpha}{\rho\alpha + 1 - \alpha}, \tag{9}$$

Computing the probabilities in (8) is slightly more complicated:

$$p_{BB}^{in} := \mathbb{P}(v \text{ receives connection from a node of color } B \mid v \in B)$$

$$= \frac{\frac{(1-r)(1-\alpha)}{\alpha\rho+1-\alpha}}{\frac{r\rho(1-\alpha)}{\alpha+\rho(1-\alpha)} + \frac{(1-r)(1-\alpha)}{\alpha\rho+1-\alpha}} = \frac{\frac{1-r}{\alpha\rho+1-\alpha}}{\frac{r\rho}{\alpha+\rho(1-\alpha)} + \frac{1-r}{\alpha\rho+1-\alpha}},$$

$$p_{BR}^{in} := \mathbb{P}(v \text{ receives connection from a node of color } R \mid v \in B)$$

$$= \frac{\frac{\rho r(1-\alpha)}{\alpha+\rho(1-\alpha)}}{\frac{r\rho(1-\alpha)}{\alpha+\rho(1-\alpha)} + \frac{(1-r)(1-\alpha)}{\alpha\rho+1-\alpha}} = \frac{\frac{\rho r}{\alpha+\rho(1-\alpha)}}{\frac{r\rho}{\alpha+\rho(1-\alpha)} + \frac{1-r}{\alpha\rho+1-\alpha}},$$

$$p_{RR}^{in} := \mathbb{P}(v \text{ receives connection from a node of color } R \mid v \in R)$$

$$= \frac{\frac{r\alpha}{\alpha+\rho(1-\alpha)}}{\frac{r\alpha}{\alpha+\rho(1-\alpha)} + \frac{\rho(1-r)\alpha}{\alpha\rho+1-\alpha}} = \frac{\frac{r}{\alpha+\rho(1-\alpha)}}{\frac{r}{\alpha+\rho(1-\alpha)} + \frac{\rho(1-r)}{\alpha\rho+1-\alpha}},$$

$$p_{RB}^{in} := \mathbb{P}(v \text{ receives connection from a node of color } B \mid v \in R)$$

$$= \frac{\frac{\rho(1-r)\alpha}{\alpha\rho+1-\alpha}}{\frac{r\alpha}{\alpha+\rho(1-\alpha)} + \frac{\rho(1-r)\alpha}{\alpha\rho+1-\alpha}} = \frac{\frac{\rho(1-r)}{\alpha\rho+1-\alpha}}{\frac{r}{\alpha+\rho(1-\alpha)} + \frac{\rho(1-r)}{\alpha\rho+1-\alpha}}. \tag{10}$$

We continue with the results on the coefficients in the exponents of the power laws that govern the degree distribution of the BPAM. It was proved in Avin et al. [4] that in the BPAM with two communities, as $N \to \infty$, the limiting degree distribution is a power law distribution with a different coefficient for each community.

Specifically, denoting by $\text{top}_k(C)$ the number of nodes of degree at least $k$ of color $C$, we have:

$$\text{top}_k(R) \sim k^{-\beta_R},$$

$$\text{top}_k(R) \sim k^{-\beta_B},$$

where $a \sim b$ means that $a$ is proportional to $b$. Moreover, Avin et al. [4] derive the closed-form expression for the power law coefficients:

$$\beta_B = 1 + \frac{1}{K_B},$$

$$\beta_R = 1 + \frac{1}{K_R}, \tag{11}$$

where

$$K_B = \frac{1}{2}\left(\frac{r\rho}{\alpha + \rho(1 - \alpha)} + \frac{1 - r}{\alpha\rho + 1 - \alpha}\right) > \frac{1}{2},$$

$$K_R = \frac{1}{2}\left(\frac{r}{\alpha + \rho(1 - \alpha)} + \frac{\rho(1 - r)}{\alpha\rho + 1 - \alpha}\right) < \frac{1}{2}. \tag{12}$$

From (11) and (12) it follows that [4]

$$\beta_R > 3 > \beta_B.$$

In Section 3, we will also need the next two propositions.

**Proposition 7.1.** *For $\beta_B$ defined in (11), (12), it holds that $\beta_B > 2$.*

PROOF. By (11) we have that $\beta_B > 2$ is equivalent to $K_B < 1$, and by (12), this is equivalent to

$$\frac{r\rho}{\alpha + \rho(1 - \alpha)} + \frac{1 - r}{\alpha\rho + 1 - \alpha} < 2. \tag{13}$$

The first fraction in (13) is smaller than one because $r < \frac{1}{2} < 1 - \alpha$. The second fraction is smaller than one because $1 - r < 1 - \alpha$ (recall the power inequality $\alpha < r$). Hence, the total left-hand side of (13) is smaller than 2. This proves the proposition. □

**Proposition 7.2.** *We have that*

$$\frac{1}{\beta_R - 1} > \frac{2}{\beta_B - 1} - 1, \tag{14}$$

*or, equivalently,*

$$2K_B - 1 < K_R. \tag{15}$$

PROOF. We will prove that $4K_B - 2K_R < 2$. Substituting the expressions (12) for $K_B$ and $K_R$, we have to prove that

$$\frac{2r\rho}{\alpha + \rho - \alpha\rho} + \frac{2(1 - r)}{\alpha\rho + 1 - \alpha} - \frac{\rho(1 - r)}{\alpha\rho + 1 - \alpha} - \frac{r}{\alpha + \rho - \alpha\rho} < 2$$

$$\Leftrightarrow \frac{r}{\alpha + \rho - \alpha\rho}(2\rho - 1) + \frac{1 - r}{\alpha\rho + 1 - \alpha}(2 - \rho) < 2.$$

Multiplying both sides of the inequality by $(\alpha\rho + 1 - \alpha)(\alpha + \rho - \alpha\rho)$, we get

$$r(2\rho - 1)(\alpha\rho + 1 - \alpha) + (1 - r)(2 - \rho)(\alpha + \rho - \alpha\rho)$$
$$< 2(\alpha + \rho - \alpha\rho)(\alpha\rho + 1 - \alpha)$$
$$\Leftrightarrow 2r\alpha\rho^2 + 2r\rho - 2r\alpha\rho - r\alpha\rho - r + r\alpha + 2\alpha + 2\rho - 2\alpha\rho$$
$$- 2r\alpha - 2r\rho + 2r\alpha\rho - \alpha\rho - \rho^2 + \alpha\rho^2 + r\alpha\rho + r\rho^2 - r\alpha\rho^2$$
$$< 2\alpha^2\rho + 2\alpha - 2\alpha^2 + 2\alpha\rho^2 + 2\rho - 2\alpha\rho - 2\alpha^2\rho^2 - 2\alpha\rho + 2\alpha^2\rho$$
$$\Leftrightarrow r\alpha\rho^2 - r - r\alpha - \rho^2 + r\rho^2 < 4\alpha^2\rho - 2\alpha^2 + \alpha\rho^2 - 2\alpha^2\rho^2$$
$$\Leftrightarrow 0 < \rho^2(1 + \alpha) + (1 - \rho)(r(1 + \alpha)(1 + \rho) - 2\alpha^2(1 - \rho))$$

Now, we know that $\rho$ and $\alpha$ are non-negative, therefore $\rho^2(1 + \alpha) \geq 0$. We also know that $\rho \leq 1$, so $1 - \rho \geq 0$. We will quickly show that $r(1 + \alpha)(1 + \rho) - 2\alpha^2(1 - \rho) \geq 0$. First, we know that $1 > r > \alpha \geq 0$. Therefore, $r(1 + \alpha) \geq 2\alpha^2 \geq 0$. Second, $1 + \rho > 1 - \rho$. We are now done. □

## 7.2 A detailed analysis of HITS

As per equation (2), the HITS update equations are defined as follows:

$$a^{(t+1)}(v) = \sum_{w:(w,v)\in E} h^{(t)}(w),$$
$$h^{(t+1)}(v) = \sum_{w:(v,w)\in E} a^{(t+1)}(w), \quad t = 0, 1, \ldots, \quad (16)$$

starting with $h^{(0)}(v) = 1$ for all $v \in V$, and ranking nodes according to the authority scores $a^{(t)}(v)$. By iterating equation (16) once, we obtain a recursion for $a^{(t+1)}(v)$ in terms of $a^{(t)}(z)$ (see equation (17) below). This recursion is central to our analysis, and is split in three terms: 1) from $z = v$; 2) $z \neq v, z \in B$; 3) $z \neq v, z \in R$. Formally, we write:

$$a^{(t+1)}(v) = \sum_{w:(w,v)\in E} \sum_{z:(w,z)\in E} a^{(t)}(z)$$
$$= d^{in}(v)a^{(t)}(v)$$
$$+ \sum_{C\in\{R,B\}} \sum_{\substack{w\in C \\ (w,v)\in E}} \sum_{\substack{z\in B \\ (w,z)\in E \\ z\neq v}} a^{(t)}(z)$$
$$+ \sum_{C\in\{R,B\}} \sum_{\substack{w\in C \\ (w,v)\in E}} \sum_{\substack{z\in R \\ (w,z)\in E, \\ z\neq v}} a^{(t)}(z), \quad t = 1, 2, \ldots. \quad (17)$$

We will derive a mean-field approximation for (17). We note that this recursion can also be thought of as path counting for backward-forward paths that start in $v$ (see Borodin et al. [9] for an example).

The first mean-field step is in approximating the fraction of red and blue in- and out-neighbors of node $w$, by the corresponding probabilities. Let $q_{CC'}$ be the probability that node $v$ of color $C$ has in-edge from node $w$ (of any color), which in turns has out-edge to node $z$ of color $C'$. Then we have

$$q_{CC'} = p_{CB}^{in} \cdot p_{BC'}^{out} + p_{CR}^{in} \cdot p_{RC'}^{out}. \quad (18)$$

Note that

$$q_{CB} + q_{CR} = 1. \quad (19)$$

The second mean-field step is in replacing $a^{(t)}(z)$ in equation (17) by the average over all vertices of the same color as $z$. This step is justified because our preferential attachment graph can be approximated by its so-called *local weak limit*, which is a continuous-time branching process (see the precise convergence result in Antunes et al. [3, Theorem 3.5]). Such processes grow exponentially in time, thus, vertex $w$ in equation (17) most likely has arrived at the end of the graph formation, so it is reasonable to assume that vertex $z$, to which $w$ connects, has average characteristics. Since the probability of connecting to $z$ is proportional to its in-degree, we replace $a^{(t)}(z)$ by its average with respect to the size-biased distribution of the in-degree. Denote by $d^{in}(v)$ the indegree of $v$. Then we obtain:

$$\overline{a^{(t)}(C)} = \frac{1}{d^{in}(C)|C|} \sum_{z\in C} d^{in}(z) \cdot a^{(t)}(z), \quad C \in \{R, B\}, \quad (20)$$

where

$$d^{in}(C) = \frac{1}{|C|} \sum_{u\in C} d^{in}(u), \quad C \in \{R, B\}, \quad (21)$$

is the average indegree of nodes of color $C$. We also denote

$$\tilde{d}_t^{in}(C) = \frac{1}{d^{in}(C)\,|C|} \sum_{u\in C} (d^{in}(u))^t, \quad \forall t \in \mathbb{N}^*, C \in \{R, B\}. \quad (22)$$

In computations below we will use the well-known results on power law distribution, namely,

$$\tilde{d}_t^{in}(C) = O_P(1), \quad \text{if } t < \beta_C - 1;$$
$$\tilde{d}_t^{in}(C) = O_P\left(n^{\frac{t}{\beta_C-1}-1}\right), \quad \text{if } t > \beta_C - 1, \quad (23)$$

where $O_P(\cdot)$ means that the big-O relation holds in probability. With these notations, the mean-field approximation, that we denote by $\approx$, of equation (17) becomes

$$a^{(t+1)}(v) \approx d^{in}(v) \cdot a^{(t)}(v)$$
$$+ d^{in}(v)(d - 1) \cdot q_{CB} \cdot \overline{a^{(t)}(B)} \quad (24)$$
$$+ d^{in}(v)(d - 1) \cdot q_{CR} \cdot \overline{a^{(t)}(R)}.$$

Now, in order to investigate the proportion of the majority (the blue vertices) in the ranking, we will iterate (24) for $t = 1, 2, \ldots$, and approximate its main term when $v$ is blue or red. As before, $|(BF)^0(v)| = 1$ for all $v \in V$. Then,

$$a^{(1)}(v) = d^{in}(v), \quad (25)$$
$$a^{(2)}(v \in B) \approx (d^{in}(v))^2 + d^{in}(v)(d - 1) \cdot q_{BB} \cdot \tilde{d}_2^{in}(B)$$
$$+ d^{in}(v)(d - 1) \cdot q_{BR} \cdot \tilde{d}_2^{in}(R), \quad (26)$$
$$a^{(2)}(v \in R) \approx (d^{in}(v))^2 + d^{in}(v)(d - 1) \cdot q_{RB} \cdot \tilde{d}_2^{in}(B)$$
$$+ d^{in}(v)(d - 1) \cdot q_{RR} \cdot \tilde{d}_2^{in}(R). \quad (27)$$

Equation (25) says that the first iteration of HITS ranks the nodes according to their indegrees. Interestingly, equations (26) and (27) show the enhancement of the majority already in the second iteration. Indeed, since $2 > \beta_B - 1$, equation (23) says that the second term in equations (26)-(27) — the mean-field contribution of the blue nodes — scales as a positive power of $N$, $\tilde{d}_2^{in}(B) = O_P\left(N^{\frac{2}{\beta_B-1}-1}\right)$,

while $\tilde{d}_2^{in}(R) = O_P(1)$ because $2 < \beta_R - 1$. When a node has a moderate degree, then its HITS approximated score after 2 iterations is dominated by the $O_P\left(N^{\frac{2}{\beta_B-1}-1}\right)$ term coming from the majority (blue) nodes. Therefore, for any vertex $v$ with bounded indegree (of the order $O(1)$), we can write

$$a^{(2)}(v \in R) \approx d^{in}(v) \cdot (d-1) \cdot MF^{(2)}(R),$$
$$a^{(2)}(v \in B) \approx d^{in}(v) \cdot (d-1) \cdot MF^{(2)}(B), \qquad (28)$$

where

$$MF^{(2)}(R) \approx q_{RB} \cdot \tilde{d}_2^{in}(B),$$
$$MF^{(2)}(B) \approx q_{BB} \cdot \tilde{d}_2^{in}(B) \qquad (29)$$

are what we call the multiplicative factors that allow to compare the HITS ranking with the degree ranking. This give more advantage to the blue vertices in the ranking than they had in the degree ranking, as we can show the following two results:

**Proposition 7.3.** The following hold true:

(1) $MF^{(2)}(B) \geq MF^{(2)}(R)$,

(2) The function $F^{(2)}(\rho) = \frac{MF^{(2)}(R,\rho)}{MF^{(2)}(R,\rho)}$ is increasing in the parameter $\rho$.

PROOF. The proof of these two results will be the base case for the induction-based proof for Propositions 3.3 and 3.4. To prove the first part, we use equation (29):

$$MF^{(2)}(B) \geq MF^{(2)}(R) \Leftrightarrow q_{BB} \geq q_{RB} \Leftrightarrow$$
$$p_{BB}^{in} \cdot p_{BB}^{out} + p_{BR}^{in} \cdot p_{RB}^{out} \geq p_{RB}^{in} \cdot p_{BB}^{out} + p_{RR}^{in} \cdot p_{RB}^{out} \Leftrightarrow \qquad (30)$$
$$\left(p_{BB}^{in} - p_{RB}^{in}\right) \cdot p_{BB}^{out} + \left(p_{BR}^{in} - p_{RR}^{in}\right) \cdot p_{RB}^{out} \geq 0.$$

We know from equation (10) that $p_{BB}^{in} + p_{BR}^{in} = 1$ and $p_{RB}^{in} + p_{RR}^{in} = 1$, and so equation (30) is equivalent to

$$\left(p_{BB}^{in} - p_{RB}^{in}\right) \cdot \left(p_{BB}^{out} - p_{RB}^{out}\right) \geq 0 \qquad (31)$$

We compute

$$p_{BB}^{out} - p_{RB}^{out} = \frac{(1-\rho^2)\alpha(1-\alpha)}{(\alpha\rho + 1 - \alpha)(\alpha + \rho(1-\alpha))} \qquad (32)$$

and

$$p_{BB}^{in} - p_{RB}^{in} = \frac{r(1-r)(\alpha + \rho(1-\alpha))(1-\rho^2)}{(r\rho u_{\alpha,\rho} + (1-r)u_{1-\alpha,\rho})(ru_{\alpha,\rho} + \rho(1-r)u_{1-\alpha,\rho})} \qquad (33)$$

where, for space brevity, we denoted by $u_{\alpha,\rho} := \alpha\rho + 1 - \alpha$ and by $u_{1-\alpha,\rho} := \rho(1-\alpha) + \alpha$.

Therefore, equation (31) is equivalent to

$$\frac{r(1-r)\alpha(1-\alpha)(1-\rho^2)^2}{(r\rho u_{\alpha,\rho} + (1-r)u_{1-\alpha,\rho})(ru_{\alpha,\rho} + \rho(1-r)u_{1-\alpha,\rho})} \qquad (34)$$

Since $\alpha, r$, and $\rho$ are smaller than 1, both the numerator and the denominator are positive.

To prove the second part, we take the expressions $q_{CC'}, p_{CC'}^{in}$, and $p_{CC'}^{out}$ for colors $C, C' \in \{R, B\}$ as functions of $\rho$, it is enough to show that $\frac{q_{RB}(\rho)}{q_{BB}(\rho)}$ is an increasing function in $\rho$. Writing out the closed-form formulas and simplifying, this reduces to showing that

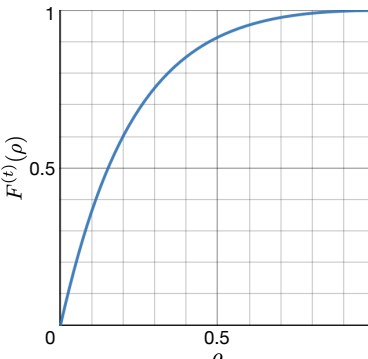

**Figure 6: Numerical plot of the $F^{(t)}(\rho)$ as a function of $\rho$.**

$$\frac{\alpha}{1-\alpha} \cdot \frac{1 - (2\alpha - r)}{2\alpha - r} \cdot \frac{\rho(1-r)(\alpha + \rho - \alpha\rho)^2 + \rho r(\alpha\rho + 1 - \alpha)^2}{(1-r)(\alpha + \rho - \alpha\rho)^2 + \rho^2 r(\alpha\rho + 1 - \alpha)^2} \qquad (35)$$

is an increasing function in $\rho$. This is easily seen by differentiating with respect to $\rho$. Note: in simplifying, we have also used the fact that $\alpha$ is the fixed point of a function

$$A(\alpha) = \frac{1}{2}\left(r + \frac{r\alpha}{\alpha + \rho - \alpha\rho} + \frac{\alpha\rho(1-r)}{\alpha\rho + 1 - \alpha}\right) \qquad (36)$$

and that $1 - \alpha$ is the fixed point of a function

$$B(\alpha) = \frac{1}{2}\left(1 - r + \frac{(1-r)(1-\alpha)}{\alpha\rho + 1 - \alpha} + \frac{r\rho(1-\alpha)}{\alpha + \rho - \alpha\rho}\right). \qquad (37)$$

These fixed points follow from the analysis in Avin et al. [4], knowing that asymptotically, the fraction of edges towards the red population (which is $\alpha$) is at equilibrium (and similarly for the fraction of edges towards the blue population, which is $1 - \alpha$). A numerical illustration of the function $F^{(2)}(\rho)$ as a function of $\rho$ can be found in Figure 6. (As we will see further on, the function $F^{(t)}(\rho)$ will be the same for $t > 2$ as well.) Finally, it is easy to notice that when $\rho = 1$, most terms simplify and we obtain $F^{(2)}(1) = 1$.

□

Moving on to a short analysis of nodes of top degree, according to the properties of power laws, the top degrees of red nodes are of order $O_P\left(N^{\frac{1}{\beta_R-1}}\right)$. We know from Proposition (7.2) that

$$\frac{1}{\beta_R - 1} > \frac{2}{\beta_B - 1} - 1, \qquad (38)$$

so after the second iteration, HITS should rank top-degree red nodes generally higher than mediocre blue nodes.

As we continue with the third iteration of HITS, we are essentially on a length 3 backward-forward path; in aggregating the number of such paths, we are required to compute $\overline{a^{(2)}(B)}$ and $\overline{a^{(2)}(R)}$,

which are the size-biased averages of equations (26) and (27). We obtain:

$$\overline{a^{(2)}(B)} = \tilde{d}_3^{in}(B) + \left(\tilde{d}_2^{in}(B)\right)^2 (d-1) \cdot q_{BB}$$
$$+ \tilde{d}_2^{in}(B) \cdot \tilde{d}_2^{in}(R) \cdot (d-1) \cdot q_{BR}$$
$$= O_P\left(N^{\frac{3}{\beta_B-1}-1}\right) + O_P\left(N^{\frac{4}{\beta_B-1}-2}\right) \qquad (39)$$
$$+ O_P\left(N^{\frac{2}{\beta_B-1}-1}\right) \cdot O_P(1)$$
$$= O_P\left(N^{\frac{3}{\beta_B-1}-1}\right),$$

$$\overline{a^{(2)}(R)} = \tilde{d}_3^{in}(R) + \tilde{d}_2^{in}(R) \cdot \tilde{d}_2^{in}(B) \cdot (d-1) \cdot q_{RB}$$
$$+ \left(\tilde{d}_2^{in}(R)\right)^2 d(d-1) \cdot q_{RR}$$
$$= O_P\left(N^{\max\{0,\frac{3}{\beta_R-1}-1\}}\right) + O_P(1) \cdot O_P\left(N^{\frac{2}{\beta_R-1}-1}\right) \quad (40)$$
$$+ O_P(1)$$
$$= O_P\left(N^{\frac{3}{\beta_B-1}-1}\right).$$

We thus note that the term $\tilde{d}_3^{in}(B)$ dominates both $\overline{a^{(2)}(B)}$ and $\overline{a^{(2)}(R)}$. We note that this term has a contribution of $q_{BB}$ in $a^{(3)}(B)$ and a contribution of $q_{RB}$ in $a^{(3)}(R)$. By the proof of Proposition 7.3 we know that $q_{BB} \geq q_{RB}$. To see exactly the closed-form approximation of this term in the HITS score of the two communities, we replace equations (39) and (40) in equation (24) for $t = 3$, obtaining

$$a^{(3)}(v \in B) = \left(d^{in}(v)\right)^3 + \left(d^{in}(v)\right)^2 (d-1) \cdot q_{BB} \cdot \tilde{d}_2^{in}(B)$$
$$+ \left(d^{in}(v)\right)^2 (d-1) \cdot q_{RB} \cdot \tilde{d}_2^{in}(R)$$
$$+ d^{in}(v)(d-1) \cdot q_{BB} \cdot \tilde{d}_3^{in}(B) + d^{in}(v)(d-1)^3 \cdot q_{BB}^2 \cdot \left(\tilde{d}_2^{in}(B)\right)^2$$
$$+ d^{in}(v)(d-1)^3 \cdot q_{BB} \cdot q_{BR} \cdot \tilde{d}_2^{in}(R) \cdot \tilde{d}_2^{in}(B)$$
$$+ d^{in}(v)(d-1) \cdot q_{BR} \cdot \tilde{d}_3^{in}(R)$$
$$+ d^{in}(v)(d-1)^3 \cdot q_{BR} \cdot q_{RR} \cdot \left(\tilde{d}_2^{in}(R)\right)^2$$
$$+ d^{in}(v)(d-1)^3 \cdot q_{BR} \cdot q_{RB} \cdot \tilde{d}_2^{in}(R) \cdot \tilde{d}_2^{in}(B),$$
$$a^{(3)}(v \in R) = \left(d^{in}(v)\right)^3 d + \left(d^{in}(v)\right)^2 (d-1) \cdot q_{RB} \cdot \tilde{d}_2^{in}(B)$$
$$+ \left(d^{in}(v)\right)^2 (d-1) \cdot q_{RR} \cdot \tilde{d}_2^{in}(R)$$
$$+ d^{in}(v)(d-1) \cdot q_{RB} \cdot \tilde{d}_3^{in}(B)$$
$$+ d^{in}(v)(d-1)^3 \cdot q_{RB} \cdot q_{BB} \cdot \left(\tilde{d}_2^{in}(B)\right)^2$$
$$+ d^{in}(v)(d-1)^3 \cdot q_{RB} \cdot q_{BR} \cdot \tilde{d}_2^{in}(R) \cdot \tilde{d}_2^{in}(B)$$
$$+ d^{in}(v)(d-1) \cdot q_{RR} \cdot \tilde{d}_3^{in}(R) + d^{in}(v)(d-1)^3 \cdot q_{RR}^2 \cdot \left(\tilde{d}_2^{in}(R)\right)^2$$
$$+ d^{in}(v)(d-1)^3 \cdot q_{RR} \cdot q_{RB} \cdot \tilde{d}_2^{in}(R) \cdot \tilde{d}_2^{in}(B).$$

Clearly, when $d^{in}(v) = O(1)$, the term $\tilde{d}_3^{in}(B)$ dominates both $a^{(3)}(v \in R)$ and $a^{(3)}(v \in B)|$, so we can write

$$a^{(3)}(v \in R) \approx d^{in}(v) \cdot (d-1) \cdot MF^{(3)}(R),$$
$$a^{(3)}(v \in B) \approx d^{in}(v) \cdot (d-1) \cdot MF^{(3)}(B), \qquad (41)$$

where

$$MF^{(3)}(R) \approx q_{RB} \cdot \tilde{d}_3^{in}(B),$$
$$MF^{(3)}(B) \approx q_{BB} \cdot \tilde{d}_3^{in}(B) \qquad (42)$$

are the multiplicative factors for $t = 3$. Thus, by the exact same proof, Proposition 7.3 can be proved for $t = 3$. A simple inductive argument will show that in all subsequent iterations of (17), the term $\tilde{d}_t^{in}(B)$ of the order $O_P\left(N^{\frac{t}{\beta_B-1}-1}\right)$, dominates $\overline{a^{(t)}(R)}$ and $\overline{a^{(t)}(B)}$, and when $d^{in}(v) = O(1)$, we can compute its coefficient in the same way as:

$$a^{(t)}(v \in R) \approx d^{in}(v) \cdot (d-1) \cdot MF^{(t)}(R),$$
$$a^{(t)}(v \in B) \approx d^{in}(v) \cdot (d-1) \cdot MF^{(t)}(B), \qquad (43)$$

where

$$MF^{(t)}(R) \approx q_{RB} \cdot \tilde{d}_t^{in}(B),$$
$$MF^{(t)}(B) \approx q_{BB} \cdot \tilde{d}_t^{in}(B) \qquad (44)$$

As we have seen before, it is now no different to generalize Proposition 7.3 for any $t > 2$.

The induction involves a few properties stemming from the recursion equations and the approximation used. We start by looking at the majority community $B$ (arguing that a similar argument goes through for community $R$):

(1) $a^{(t)}(v \in B)$ is a polynomial in $d^{in}(v)$ of degree $t$, with a leading coefficient equal to $d$;
(2) The term $d_t^{in}(B)$ is the dominant term (asymptotically in $N$) in $a^{(t)}(v \in B)$, as a coefficient to $d^{in}(v)$.

The first point is easy to see by induction, knowing our base case from equations (26) and (27) and the recursion equation (24) (which is a linear equation in $d^{in}(v)$). For the second point, we'll use the mean-field equations and the induction hypothesis (assumed true for $t - 1$ with the goal of showing it for $t$). When computing $\overline{a^{(t-1)}(B)}$, we essentially average $a^{(t-1)}(z)$ over all $z \in B$. Since from the induction hypothesis we know that $a^{(t-1)}(v \in B)$ is a polynomial in $d^{in}(v)$ of degree $t - 1$ with a leading coefficient equal to $d$, we will show that the dominant term of $\overline{a^{(t-1)}(B)}$ is $d_t^{in}(B)$. First of all, clearly this term exists with coefficient $d$ from our previous remark. Secondly, we need to show that it is in fact the dominant term. Since from the induction hypothesis the term $d_{t-1}^{in}(B)$ is the dominant term (asymptotically in $N$) in $a^{(t-1)}(v \in B)$, as a coefficient to $d^{in}(v)$, this term will turn into $d_{t-1}^{in}(B) \cdot d_2^{in}(B)$ in the averaging process (with some coefficient). Now, in comparing $d_{t-1}^{in}(B) \cdot d_2^{in}(B)$ and $d_t^{in}(B)$, $d_t^{in}(B)$ clearly dominates, since

$$\frac{t}{\beta_B-1} - 1 > \frac{t-1}{\beta_B-1} - 1 + \frac{2}{\beta_B-1} - 1 \Leftrightarrow \beta_B > 2, \qquad (45)$$

which we know to be true. Finally, no other term in $\overline{a^{(t-1)}(B)}$ is competitive by the same argument. By a similar argument, $d_t^{in}(R)$ is the dominant term in $\overline{a^{(t-1)}(R)}$. Thus, $d_t^{in}(B)$ is the dominant term in $a^{(t)}(v \in B)$, as the summand $a^{(t-1)}(B)$ in the approximation contributes $\underline{d_{t-1}^{in}(B)}$ as a dominant term (clearly dominated), the summand $\overline{a^{(t-1)}(B)}$ contributes $d_t^{in}(B)$ as a dominant term, and the summand $\overline{a^{(t-1)}(R)}$ contributes $d_t^{in}(R)$ as a dominant term (clearly dominated as $\beta_B < \beta_R$). Thus, this shows the second part, that the term $d_t^{in}(B)$ is the dominant term (asymptotically in $N$) in $a^{(t)}(v \in B)$, as a coefficient to $d^{in}(v)$ (coming from $d^{in}(v)$ multiplied by $\overline{a^{(t-1)}(B)}$ in the mean-field approximation).

We finalize our analysis by looking at the top red vertices with degree $O_P\left(N^{\frac{1}{\beta_R-1}}\right)$. In iteration $t > 2$, the largest contribution of their degree is $O_P\left(N^{\frac{t}{\beta_R-1}}\right)$, while the largest competing term comes from the mean-field contribution of the blue vertices in $a^{(t-1)}(v)$, so this term is $(d^{in}(v))^2 \cdot \overline{a^{(t-2)}(B)} = O_P\left(N^{\frac{2}{\beta_R-1}+\frac{t-1}{\beta_B-1}-1}\right)$. When $t = 3$, the degree term is still of a larger order of magnitude due to (38). However, in subsequent iterations, since

$$\frac{1}{\beta_B - 1} > \frac{1}{\beta_R - 1},$$

the mean-field contribution of the blue vertices grows faster, has an increasing share of the BF-paths, and at the same time, contributes with a smaller factor $q_{BR} \leq q_{BB}$. This explains the fact that the majority vertices are increasingly enhanced in subsequent iterations of HITS, as the backward-forward paths continue.

## 7.3 A detailed analysis of randomized HITS

In this section, we detail a short argument for why randomized HITS closely follows the degree ranking in BPAM. We recall the iterative processed defining randomized HITS from equation (6), which we transpose and rewrite for ease of notation (noting that $a(\cdot)$ and $h(\cdot)$ are row vectors and now $\overrightarrow{\mathbb{1}}$ defines the row of ones):

$$a^{(t+1)} = \epsilon \cdot \overrightarrow{\mathbb{1}} + (1 - \epsilon) \cdot h^{(t)} \cdot A_{row},$$
$$h^{(t+1)} = \epsilon \cdot \overrightarrow{\mathbb{1}} + (1 - \epsilon) \cdot a^{(t+1)} \cdot A_{col}^T, \tag{46}$$

where $\overrightarrow{\mathbb{1}}$ is the all-ones vector, $A_{row}$ and $A_{col}$ are the row- and column-stochastic versions of the adjacency matrix $A$, respectively.

We quickly note that $A_{row}$ is essentially normalizing the adjacency matrix by the constant outdegree of the BPAM. Equivalently, $A_{row}$ is the transition matrix for taking a 'forward' step from a node, following their outdegree. Similarly, $A_{col}^T$ is essentially normalizing the adjacency matrix by the indegree of the BPAM, equivalent to being the transition matrix for taking a 'backward' step from a node, following their indegree. Iterating the recursion of authority scores from equation (46), we get:

$$a^{(t+1)} = \epsilon \cdot \overrightarrow{\mathbb{1}} + \epsilon(1-\epsilon) \cdot \overrightarrow{\mathbb{1}} \cdot A_{row} + (1-\epsilon)^2 \cdot a^{(t)} \cdot A_{col}^T A_{row} \tag{47}$$

Equation (47) is similar to the Pagerank equation (1), with transition matrix $A_{col}^T A_{row}$ (the backward-forward matrix) and damping

factor $(1 - \epsilon)^2$. We observe that

$$\left(\overrightarrow{\mathbb{1}} \cdot A_{row}\right)(v) = \sum_{w=1}^{N} (A_{row})_{wv} = \sum_{w:(w,v)\in E} \frac{1}{d^{out}(w)}. \tag{48}$$

Notice that in our BPAM with constant out-degree $d$, we get

$$\overrightarrow{\mathbb{1}} \cdot A_{row} = \frac{1}{d} \mathbf{d}^{in}, \tag{49}$$

where $\mathbf{d}^{in}$ is the row vector of indegrees

$$\mathbf{d}^{in} = \left(d_1^{in}, d_2^{in}, \cdots, d_N^{in}\right). \tag{50}$$

Iterating equation (47) over $t$ in BPAM, we obtain:

$$a^{(t+1)} = (1-\epsilon)^{2t} \cdot a^{(1)} \cdot \left(A_{col}^T A_{row}\right)^t$$
$$+ \left(\frac{\epsilon(1-\epsilon)}{d} \cdot \mathbf{d}^{in} + \epsilon \cdot \overrightarrow{\mathbb{1}}\right) \cdot \sum_{k=0}^{t-1} (1-\epsilon)^{2k} \left(A_{col}^T A_{row}\right)^k. \tag{51}$$

We note that

$$a^{(1)} = \frac{1-\epsilon}{d} \cdot \mathbf{d}^{in} + \epsilon \cdot \overrightarrow{\mathbb{1}}, \tag{52}$$

yielding

$$a^{(t+1)} = (1-\epsilon)^{2t} \cdot \left(\frac{1-\epsilon}{d} \cdot \mathbf{d}^{in} + \epsilon \cdot \overrightarrow{\mathbb{1}}\right) \cdot \left(A_{col}^T A_{row}\right)^t$$
$$+ \frac{\epsilon(1-\epsilon)}{d} \cdot \mathbf{d}^{in} + \epsilon \cdot \overrightarrow{\mathbb{1}} \tag{53}$$
$$+ \left(\frac{\epsilon(1-\epsilon)}{d} \cdot \mathbf{d}^{in} + \epsilon \cdot \overrightarrow{\mathbb{1}}\right) \cdot \sum_{k=1}^{t-1} (1-\epsilon)^{2k} \left(A_{col}^T A_{row}\right)^k.$$

To get more insight into further iterations, we compute

$$\left(A_{col}^T A_{row}\right)_{zv} = \sum_{w=1}^{N} \frac{A_{wz}}{d^{in}(z)} \cdot \frac{A_{wv}}{d}.$$

Now, we approximate $A_{wv}$ by its average conditioned on degrees:

$$A_{wv} \approx \frac{d^{in}(v)d}{Nd} = \frac{d^{in}(v)}{N}.$$

Substituting this, we derive

$$\left(\overrightarrow{\mathbb{1}} \cdot A_{col}^T A_{row}\right)(v) = \sum_{z=1}^{N} 1 \cdot \left(\sum_{w=1}^{N} \frac{A_{wz}}{d^{in}(z)} \cdot \frac{A_{wv}}{d}\right)$$
$$\approx \sum_{w=1}^{N} \sum_{z=1}^{N} \frac{d^{in}(z)}{d^{in}(z)N} \cdot \frac{d^{in}(v)}{Nd} = \frac{d^{in}(v)}{d}.$$

Also, we notice that in further iterations,

$$\left(\mathbf{d}^{in} \cdot A_{col}^T A_{row}\right)(v) = \sum_{z=1}^{N} d^{in}(z) \cdot \left(\sum_{w=1}^{N} \frac{A_{wz}}{d^{in}(z)} \cdot \frac{A_{wv}}{d}\right)$$
$$\approx \sum_{w=1}^{N} \sum_{z=1}^{N} d^{in}(z) \frac{d^{in}(z)}{d^{in}(z)N} \cdot \frac{d^{in}(v)}{Nd} = d^{in}(v).$$

From this and equation (53), we see that in the mean-field approximation, $a^{(t+1)}$ has the term $\epsilon \cdot \overrightarrow{\mathbb{1}}$, and the rest of the terms proportional to $\mathbf{d}^{in}$. We conclude that in BPAM, randomized HITS ranks approximately by the indegree.

In a real-world dataset, the outdegrees are not constant, so, for instance, $a^{(1)}(\cdot)$ ranks nodes by the right-hand side of equation (48), and all subsequent iterations will have such term as well. Then, like in PageRank, it is beneficial to receive edges from nodes of small out-degree, but unlike in PageRank, the contribution of high in-degree neighbors is counterbalanced thanks to the division by indegrees in $A_{col}^T$. If minority nodes tend to receive edges from nodes of lower outdegree, then they will benefit in the ranking produced, achieving higher ranks.

## 7.4 Subspace HITS: an analysis of various eigenvectors

We experiment with a various number of eigenvectors and aggregations functions $f$ in Figures 7–9. Panels (b) and (c) show the percentage of minority present at each rank and above, choosing the first $x$ eigenvectors of the $A^T A$ matrix, as $x$ varies between 1 and 10, and aggregating them using $f(\lambda) = 1$ (b) and $f(\lambda) = \lambda^2$ (c). We note that for the larger datasets (DBLP and Instagram), the choice of $f$ between $f(\lambda) = 1$ and $f(\lambda) = \lambda^2$ does not change the ranking, as the gap between different values of the authority scores obtained is larger than the values of the eigenvalues squared. The trends are quite different, depending on the number of eigenvectors chosen, noting that each datasets seems to have a different 'optimum' in terms of fairness. For example, for the DBLP dataset, choosing 5 eigenvectors seems to have the best fairness improvement (and even better than the degree ranking, as Figure 8 (a) shows); choosing any other number presents no clear pattern. Cucuringu and Mahoney [12] present an empirical analysis of when a minority group gets captured in a lower-order eigenvector, using the *inverse participation ratio* as a measure of a score over eigendirections describing how well-captured a community is in a given eigendirection. They note that a community may appear well-represented in various lower-order eigenvectors. We conclude that using subspace HITS may be in some cases beneficial, but without a stable and consistent pattern for how many eigenvectors might improve fairness for a minority group.

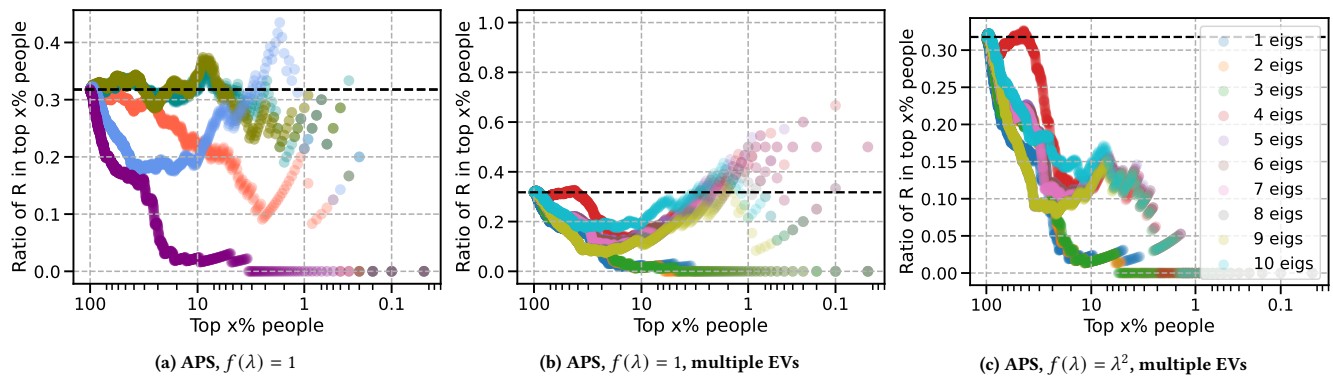

(a) APS, $f(\lambda) = 1$

(b) APS, $f(\lambda) = 1$, multiple EVs

(c) APS, $f(\lambda) = \lambda^2$, multiple EVs

**Figure 7: Representation of the minority group R in the ranking of the nodes based on degree (orange), HITS (purple), and Pagerank (blue), for the APS dataset, using 6 eigenvectors and $f(\lambda) = 1$ (a). Figures (b) and (c) show the representation of the minority group when using a varied number of eigenvectors for $f(\lambda) = 1$ (b) and $f(\lambda) = \lambda^2$ (c).**

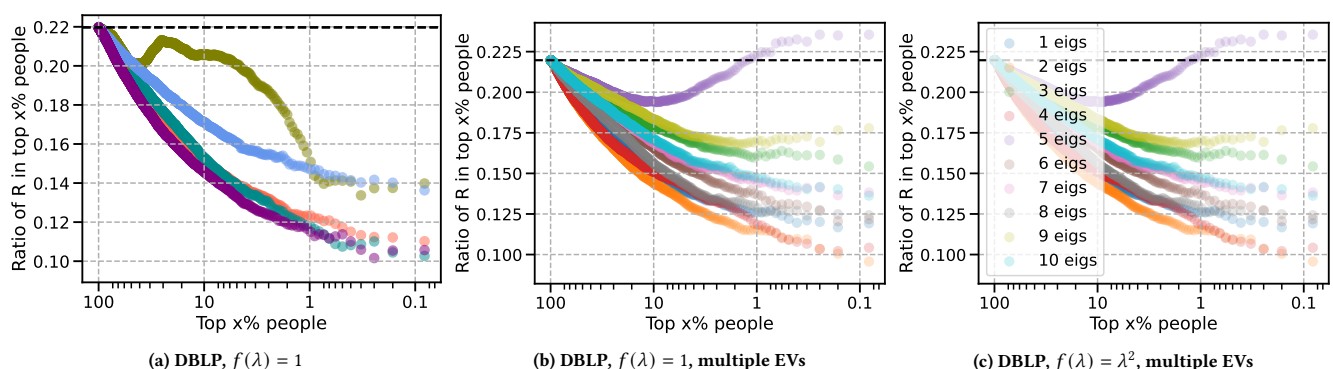

(a) DBLP, $f(\lambda) = 1$

(b) DBLP, $f(\lambda) = 1$, multiple EVs

(c) DBLP, $f(\lambda) = \lambda^2$, multiple EVs

**Figure 8: Representation of the minority group R in the ranking of the nodes based on degree (orange), HITS (purple), and Pagerank (blue), for the DBLP dataset, using 10 eigenvectors and $f(\lambda) = 1$ (a). Figures (b) and (c) show the representation of the minority group when using a varied number of eigenvectors for $f(\lambda) = 1$ (b) and $f(\lambda) = \lambda^2$ (c).**

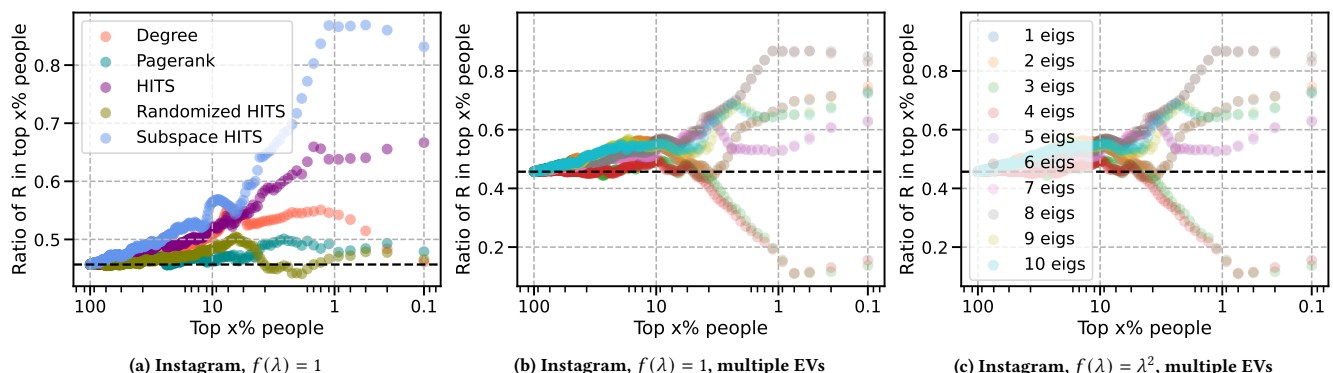

(a) Instagram, $f(\lambda) = 1$

(b) Instagram, $f(\lambda) = 1$, multiple EVs

(c) Instagram, $f(\lambda) = \lambda^2$, multiple EVs

**Figure 9: Representation of the minority group R in the ranking of the nodes based on degree (orange), HITS (purple), and Pagerank (blue), for the Instagram dataset, using 6 eigenvectors and $f(\lambda) = 1$ (a). Figures (b) and (c) show the representation of the minority group when using a varied number of eigenvectors for $f(\lambda) = 1$ (b) and $f(\lambda) = \lambda^2$ (c).**

