# OpenReview forum: "Fairness in link analysis ranking algorithms"
_ACM.org/TheWebConf/2024/Conference — TheWebConf24_

### Official Review · Reviewer_T3az · 2023-11-12

**Novelty:** 5
**Technical Quality:** 5

**Review:**

This paper extensively analyzes inherent biases in existing ranking algorithms, specifically Pagerank and HITS. The phenomenon of ranking bias refers to the underrepresentation of nodes in minority groups in the rankings generated by these algorithms. This study reveals that while Pagerank tends to reflect degree rankings, HITS amplifies the bias. The authors theoretically analyze the root causes of the bias within HITS, attributing it to the homophily of the network structure. Additionally, they present a potential approach to mitigate this issue from randomization.

This work addresses an important problem caused by popular ranking algorithms both theoretically and empirically. However, some concerns are:

- While homophily can be one of the significant factors contributing to bias, other structural properties of networks, such as assortativity, transitivity, clustering coefficient, and community structure, can be considered as potential additional factors of bias. A more comprehensive examination of these factors is recommended for a thorough understanding of this issue.
- The authors used only three real-world datasets to confirm their theoretical analysis. However, this limited dataset may not fully validate their conclusions. More extensive experiments using a wider range of real-world datasets are recommended for a more comprehensive evaluation.
- The minority ratio of APS dataset is inconsistent. In Table 1, the ratio is reported as 31.7% while in the text, it is mentioned that CSM consists 37.5% of the paper topics.
- Alternative measures for network homophily, such as the Newman assortativity index and asymmetric homophily index, which are mentioned but not explored, are worth examining.

**Questions:**

- Please find the concerns mentioned above.
- This paper considers two groups - majority and minority, in BPAM and real-world datasets. Is it possible to extend the analysis by considering multiple groups with varying levels of minority/majority?
- Are techniques commonly employed to tackle long-tail problems, such as reweighting or rebalancing, worth considering for mitigating bias in HITS?

**Reviewer Confidence:**

3: The reviewer is confident but not certain that the evaluation is correct

**Scope:**

4: The work is relevant to the Web and to the track, and is of broad interest to the community

---

### Official Review · Reviewer_jyvw · 2023-11-20

**Novelty:** 3
**Technical Quality:** 5

**Review:**

In this paper, the authors study how PageRank and HITS behave in terms of inequality between different social groups. In particular, they aim to theoretically explain when and why the ranking obtained by these two algorithms reduces, mirrors, or amplifies the bias that is observed by the degree distribution of different communities, with the assumption that "the degree ranking is a baseline for social capital inequality".

The analysis is conducted on synthetic data coming from the so-called Biased Preferential Attachment Model, where a starting network is populated by adding at each step one node and one directed edge. The authors state that the model is repeated $d$ times so that every node has an out-degree of $d$, but how this is done is vague since the model does not provide the possibility to add one directed edge from an existing node. Moreover, it is not made explicit if at time $0$ the network is empty or not.

The authors implement PageRank by using a value of 0.15 as restart probability and a uniform teleportation vector, as these are the default choices that usually one can find in the literature. Actually, different values of restart probability can highly impact the ranking produced by the algorithm, the same holding for different choices of the teleportation vector. It would be interesting to look at what happens for different values of these two parameters.

As a possible solution, the authors propose to add a restart probability to the HITS algorithm, obtaining what appears to be empirically the fairest algorithm. This can be a hint that different choices of restart probability and teleportation vector could possibly serve as another step toward fairness in link analysis, even if PageRank already seems to behave better than HITS. A Personalized PageRank with a vector giving more probability to teleport on nodes from the minority group is just one example of the different approaches that can be followed.

PROS
- The paper is in general well-written
- The issue is described extensively, also referring to previous literature on the topic
- The analysis is performed both theoretically and empirically

CONS
- The technical analysis and proofs are not so easy to read and most of them are in the appendix of the paper
- The code for reproducibility and comparisons does not seem to be provided
---
I've read the rebuttal of the authors to my review and those of the other reviewers, and I've updated my score accordingly.

**Questions:**

1. Could you discuss more on the choices of the parameter $n$ and $d$ for the BPAM simulations? Why only one fixed value for both parameters?
2. Even if PageRank seems to behave better than HITS, have you also thought of any solution that could improve it in terms of fairness?

**Ethics Review Description:**

No ethical issue was observed.

**Reviewer Confidence:**

2: The reviewer is willing to defend the evaluation, but it is likely that the reviewer did not understand parts of the paper

**Scope:**

3: The work is somewhat relevant to the Web and to the track, and is of narrow interest to a sub-community

---

### Official Review · Reviewer_8e2b · 2023-11-24

**Novelty:** 5
**Technical Quality:** 5

**Review:**

**1. Summary**

This paper examines bias in network algorithms utilized in link analysis ranking. By doing so, the authors demonstrated that PageRank and HITS yield significantly different outcomes. PageRank corrects the degree bias for top-ranked nodes, while randomization in HITS improves fairness. The authors also offer robust theoretical analysis and empirical evidence.

**2. Pros**

(S1) Important and Timely Problem

(S2) Strong Theoretical Analysis on Bias

(S3) Extensive Empirical Validation

**3. Cons**

(W1) Unclear Implications and Validity of Analysis/Findings

Although I appreciate the authors' strong theoretical analysis and extensive empirical validation, it remains unclear whether PageRank and HITS are still valid in modern ranking systems. It is evident that current ranking systems were designed with the philosophy of these methods in mind, making this analysis useful as a cornerstone for future research. However, it is essential to clarify the connection of the results with the current complex deep-learning-based ranking systems. In this context, it would be valuable to discuss the validity of the authors' findings in a graph-neural-networks (GNNs)-based ranking system since GNNs are significantly related to the two ranking algorithms.

(W2) Insufficient Literature Review

The fairness-aware learning-to-rank methods discussed in Section 2.2 may be considered outdated. Thoroughly reviewing recent studies, including preprocessing methods, is imperative. Although [1] does not directly relate to ranking systems, its inclusion is necessary because it addresses fairness in graph mining.

[1] Dong, Yushun, et al. "Fairness in graph mining: A survey." IEEE Transactions on Knowledge and Data Engineering (2023).

=====POST-REBUTTAL COMMENTS========

I have read the authors' responses and appreciate their efforts to address my concerns.
I still lean towards accepting this paper, and therefore, I will maintain my current score.

**Questions:**

Please see the detailed reviews and questions above.

**Reviewer Confidence:**

3: The reviewer is confident but not certain that the evaluation is correct

**Scope:**

3: The work is somewhat relevant to the Web and to the track, and is of narrow interest to a sub-community

---

### Official Review · Reviewer_k7Us · 2023-11-24

**Novelty:** 4
**Technical Quality:** 4

**Review:**

This work studies the "fairness" of two famous centralities --- PageRank and HITS. Roughly speaking, a centrality (or its induced ranking) is considered unfair if it exacerbates the role of degrees or the fact of belonging to a highly-connected community. The work analyses how fair PageRank and HITS are under a certain graph generation model called BPAM (Biased Preferential Attachment Model). The main claims are that HITS is unfair, while PageRank is (somewhat) fair; the work also analyses variants of HITS (proposed in previous work) that could potentially make it fairer. The findings are supported by experiments.

The question addressed by the work seems interesting in principle. There are however some crucial issues on which I have doubts.

1) First, in some sense it is well known that HITS and PageRank are respectively "unfair" and "fair". Take a graph containing, say, just one arc $(u,v)$. The HITS authority score will be 0 for all vertices and 1 for $v$, and the HITS hub score will be 0 for all vertices except 1 for $u$. If $G$ is the union of a 6-clique and two 2-cliques, the authority scores are 1 for every vertex in the 6-clique and $0,0041$ for every other vertex; in general, I guess in these graphs the authority will be proportional to the *squared* degrees (look at $AA^T$). In any case, this is a well-known feature of HITS. For PageRank the opposite tends to hold, in the sense that the PageRank score of a vertex $u$ can be written as $P(u) = (1-a)/n * (1 + a \sum_{v -> u} 1/out(u) + a^2 ...)$, and thus the indegree of $u$ contributes to $u$'s score with a weight that depends on the outdegrees of the vertices. (This is not completely precise, since the same incoming vertices could reappear in the subsequent terms of the infinite sum). In the example graphs above, for instance, all vertices have the same PageRank score $1/n$, regardless of their indegree. This is very well-known, too.

2) Thus, the present work basically studies HITS and PageRank under BPAM. All claims about "fairness" or not are thus basically claims about BPAM, in some sense. I am not totally sure what the meaning of this is.

3) The results on the fairness of the HITS algorithm (Theorem 3.2) seem rather weak. The theorem just says that the average authority score of the minority vertices is at most the average authority score of the minority vertices (these are defined by the BPAM model). That is a very weak claim; the two average scores might well be equal. The second part of the claim says that the ratio between the authority score of two fixed majority and minority verticesof "similar degree" (what does that mean?) is an increasing function of the homophily parameter. That is a weak result, too -- the function may be very, very slow.

4) There are several obscure points in the main theoretical result (Theorem 3.2), including undefined/vague terms, as well that all quantities should be in expectation over the graph drawn from the BPAM model.

5) There is no theoretical analysis for PageRank.

Specific comments:

- Theorem 3.2. Something's odd here. The claim is about a graph obtained from a randomized attachment model, that is, a random graph. Yet, there seems to be no trace of this in the claim--- there is no "expected value", but only an "average value" which however is an average over a set of vertices.
- Theorem 3.2. What is the "degree class" of a vertex? I do not see it defined anywhere.
- Theorem 3.2. What does "nodes of "similar degree" mean? This is not formal at all.
- I do not understand what $MF^{(t)}$ is. Above Proposition 3.3 it is defined as "the multiplicative factor that allow to compare the HITS ranking with the degree ranking". What does this mean? Note that
- Proposition 5.1. If $A(\epsilon,d,t)$ is a constant in $t$ why is it written as a function of $t$? Or is it perhaps *bounded* from above by a constant in $t$?

**Questions:**

See my points above.

**Ethics Review Description:**

-

**Reviewer Confidence:**

3: The reviewer is confident but not certain that the evaluation is correct

**Scope:**

4: The work is relevant to the Web and to the track, and is of broad interest to the community

---

### Official Review · Reviewer_tVrw · 2023-11-27

**Novelty:** 3
**Technical Quality:** 5

**Review:**

Summary: This paper provides a comparison of major methods for measuring the importance of vertices in a network: the page rank method and the HITS methods.

The analysis is carried out by considering the biased preferential attachment model. And the authors provide some theoretical evidence based on mean field analysis as well as experimental findings.
The claims are that: Page Rank often mirrors degree distribution.
The HITS algorithm can increase the bias for nodes with the same degree but from different groups.
After showing HITS increases bias, the authors consider two variations of it:
Randomized HIT in which the walks restart to a u.a.r. chosen node (like page rank ) can reduce bias , Subspace HITS: where in addition to the principal eigenvectors of AA^T, they consider a few extra dimensions,  does not help with bias.

The claims about Page Rank do not have any theoretical evidence, chapter 3.3 of the paper is devoted to theoretical results showing that HITS increases bias. The part in which the authors claim that randomized HIT has lower bias than HITS in proven in section 5.

Pros: authors consider an interesting problem and provide theoretical and experimental evidence for some of the claims.

Cons: (1 )while the authors claim that they are comparing HITS and PageRank, I couldn't find any theoretical results about page rank.
(2) The results are interesting but raise a few questions (which I will list later) which addressing could make the discussion of the paper complete. (3) some results are presented without rigor for example theorem 3.2 (2) it is written {\it nodes with similar degrees}. Proposition 5.1 A(d,\epsilon, t) what is d?

Minor notes for authors:

In the introduction of HITS algorithm what are a^(0), h^(0). Also it seems like in the rest of the paper the focus is on a please provide some explanation or note here so the reader does not get confused.


-----

I have read the responses in the rebuttal phase.  Most of my questions were answered and in response to CONS (1) authors accepted to make some changes.

I am changing my vote accordingly.

**Questions:**

(1) In theorem 3.2 where it is shown that the ratio of average authorities is increasing with time, can you say anything about the rate of this growth?

(2) In proposition 3.4, is F just a function of \rho? why is it independent of r? One would think that since r contributes to the size of a community, it may play a role here

(3) In proposition 5.1 what is d in A(d,\epsilon, t)? and when you say it is a constant in t does it mean that it is independent of it?

---
Above questions were answered by the authors in rebuttal phase, and my concerns have been addressed.

**Ethics Review Description:**

no issue

**Reviewer Confidence:**

3: The reviewer is confident but not certain that the evaluation is correct

**Scope:**

3: The work is somewhat relevant to the Web and to the track, and is of narrow interest to a sub-community

---

### Decision · Program_Chairs · 2024-01-22

**Decision:**

Accept

**Comment:**

This paper investigates the possibility of exacerbation of existing biases when link analysis algorithms (e.g., HITS and PageRank) are run in homophilic networks. Specifically, the authors compare the rankings of nodes from a minority group against their degrees (which correspond to the extant bias in the network itself), in both simulated [biased preferential attachment, BPAM] and real network data. They also look ways to mitigate these effects in HITS.

 The reviewers find the paper's topic important (even if the models that they are analyzing are from some time ago, the questions are increasingly recognized as critical ones to study). They are also positive on the strength of the clarity of the writing and the presentation, and the combination of theoretical and empirical analysis. As multiple reviewers point out (and the authors acknowledge in response), the submission seems to present itself as providing more theoretical analysis of PageRank than in fact is present in the text; the theoretical parts of this paper are fundamentally about HITS and BPAM (though PageRank appears in the empirical sections). There is a similar sense that the paper could more clearly state its contributions with respect to BPAM (as opposed to general graphs), and better situate itself in the literature. Other concerns include limitations of the model (is homophily the only dimension of interest?) and the relatively small set of real-world datasets examined. The authors plan to revise their presentation to make the scope of their contributions more clear in these regards.

 In places, the reviewers point out issues with the precision of the theorem statements and the proofs. For example, multiple reviewers point out that, e.g., Theorem 3.2 only says that the bias is in the expected direction (against the minority group), and does not establish anything about how much (additional) bias there is. The authors responded to this issue in their replies to the reviewers, satisfying some (but not all) of the concerns through their reference to an extension of their arguments regarding the magnitude of the effects.